# TimeLAVA: Learning-Agnostic Valuation for Time Series Data

**Wenqin Liu**[1]  **Weizhi Quan**[1]  **Aoqi Zuo**[2]  **Erdun Gao**[3][4]  **Vu Nguyen**[5]
**Dino Sejdinovic**[3][4]  **Howard Bondell**[1]  **Mingming Gong**[1][6]

## Abstract

Data valuation quantifies the intrinsic quality of individual samples to enable principled data curation, quality control, and robust learning. For time series in critical domains such as healthcare, finance, and industrial monitoring, effective valuation methods are essential yet fundamentally lacking. Existing approaches are either model-dependent, limiting their generalizability, or designed for i.i.d. data and thus fail to capture temporal dependencies, multi-scale patterns, and non-stationary dynamics inherent to sequential data. We introduce TIMELAVA, a learning-agnostic framework that values temporal segments by their marginal contribution to minimizing distributional discrepancy between evaluated and reference data. At its core is a novel Selective Wavelet-based Wasserstein ($\mathcal{W}_{\mathrm{SW}}$) discrepancy combining multi-scale wavelet transforms for temporal localization with unbalanced optimal transport for robustness to distributional shifts. Segment values are efficiently computed via sensitivity analysis without requiring model training and aggregated into point-wise scores. We provide theoretical guarantees linking valuation to model-agnostic generalization and prove bounded sensitivity to outlier contamination. Extensive experiments across anomaly detection, data pruning, and label noise detection demonstrate that TIMELAVA produces significantly more informative value scores than existing methods on diverse real-world datasets. The code is available at https://github.com/lww28/TimeLAVA.

## 1. Introduction

Time series data forms the backbone of modern decision-making systems. In intensive care units, continuous monitoring of physiological signals informs life-critical interventions (Celi et al., 2013; Johnson et al., 2016); in financial markets, millisecond-level price fluctuations guide trading strategies (Franses & Van Dijk, 2000; Sezer et al., 2020); in industrial IoT systems, sensor streams enable predictive maintenance (Carvalho et al., 2019). Across these domains, identifying influential temporal segments, including both informative patterns and anomalies, is essential for reliable modeling (Hamilton, 1994; Pang et al., 2021). Yet misidentification has tangible consequences: models that fail under distribution shift, spurious anomaly alerts that erode trust, or labeling budgets wasted on uninformative data. This raises a central question: *how can we quantify the intrinsic value of temporal segments in time series data?*

**Motivation.** Despite its importance, time series data valuation remains fundamentally underexplored. Existing methods, predominantly designed for i.i.d. settings, prove ill-suited for these challenges. Model-dependent approaches like Influence Functions (Koh & Liang, 2017) quantify how individual samples affect model predictions. While Time-Inf (Zhang et al., 2025) extends this to time series through temporal blocking, it remains fundamentally model-specific and assumes stationarity. Learning-agnostic methods using Optimal Transport (OT), such as LAVA (Just et al., 2023) and SAVA (Kessler et al., 2025) for i.i.d. data, define value through distributional similarity but are fundamentally unsuited for time series due to temporal dependencies and non-stationary dynamics. This leaves a critical gap: *the absence of a learning-agnostic valuation framework capable of capturing the complex, evolving dynamics inherent to sequential data.*

**Challenges.** Valuing time series data presents a distinct set of challenges stemming from its sequential nature. Observations in sequential data are linked through *temporal dependencies*, where value depends not only on their content but also on the surrounding context and the order in which they appear. This intrinsic structure is further complicated by the fact that real-world time series are often *non-stationary*: regime shifts, seasonal effects, and evolving dynamics can

---

[1]School of Mathematics and Statistics, The University of Melbourne [2]School of Mathematics and Statistics, University of Sydney [3]Responsible AI Research Centre, Australian Institute for Machine Learning [4]School of Mathematical Sciences, Adelaide University [5]Amazon [6]Department of Machine Learning, MBZUAI. Correspondence to: Erdun Gao <erdun.gao@adelaide.edu.au>.

*Proceedings of the 43rd International Conference on Machine Learning*, Seoul, South Korea. PMLR 306, 2026. Copyright 2026 by the author(s).

alter statistical properties over time. Furthermore, valuable information is often distributed across *multiple temporal scales*, with long-term cyclic trends and short-lived transient events contributing in complementary ways. In addition, practical constraints demand that any valuation method be able to process *large-scale*, high-frequency datasets efficiently while still preserving fine-grained insight.

**Contributions.** To address these challenges, we introduce TIMELAVA, a learning-agnostic framework that values time series segments by quantifying their marginal contribution to minimizing distributional discrepancy between evaluated and reference data. Our approach is built on a key insight: time series valuation is fundamentally a *selective matching* problem, where segments that cannot be well-matched to the reference distribution should remain unmatched rather than forced into poor alignments. To this end, we propose the Selective Wavelet-based Wasserstein ($\mathcal{W}_{\text{SW}}$) discrepancy, which integrates (1) *unbalanced optimal transport* to relax mass conservation constraints, enabling selective matching that naturally surfaces distributional outliers, and (2) *wavelet-based ground metrics* to capture localized temporal patterns across multiple scales. Segment values are efficiently derived via sensitivity analysis of the $\mathcal{W}_{\text{SW}}$ dual formulation, requiring no model retraining. We provide theoretical guarantees including bounded sensitivity to outlier contamination (Theorem 5.1) and connections to model-agnostic generalization (Theorem 5.3), and demonstrate consistent improvements over strong baselines on anomaly detection, data pruning, and label noise detection across healthcare, finance, and IoT benchmarks.

## 2. Related Work

**Model-dependent Valuation.** Model-dependent methods quantify data value relative to specific predictive models. Leave-one-out retraining provides direct performance measurements but is computationally prohibitive for large datasets. Shapley value methods (Jia et al., 2019b; Kwon & Zou, 2021) attribute value through marginal contributions across data subsets; however, exact computation requires exponential evaluations. Influence Functions (Hampel, 1974; Koh & Liang, 2017) estimate how reweighting a data point perturbs model parameters without retraining, with TIME-INF (Zhang et al., 2025) extending this to time series through temporal blocking. However, these approaches remain tied to specific model architectures and training procedures, limiting their generalizability across tasks.

**Learning-Agnostic Valuation.** Learning-agnostic approaches aim to quantify intrinsic data value without pre-specifying a learning algorithm. The most prominent paradigm uses distributional similarity via OT: LAVA (Just et al., 2023) and its memory-efficient variant SAVA (Kessler et al., 2025) to value i.i.d. data points by their contribution

to reducing the Wasserstein distance between training and validation distributions, providing efficient model-agnostic estimates that generalize across tasks. However, these frameworks assume independent samples, making them fundamentally unsuited for time series where value depends on temporal context and evolving non-stationary dynamics.

**Optimal Transport for Time Series.** While OT provides a principled framework for comparing distributions (Villani et al., 2008; Peyré & Cuturi, 2019), applying it to sequential data requires careful cost function design. Classical approaches like Dynamic Time Warping (Berndt & Clifford, 1994) align sequences but lack the distributional perspective of OT. Unbalanced OT (Chizat et al., 2018; Séjourné et al., 2019) relaxes strict mass conservation through divergence regularization, providing robustness to outliers. Recent work has applied spectral-domain UOT for imputation (Wang et al., 2025) to handle regime shifts. However, frequency-domain approaches sacrifice temporal localization, remaining insensitive to transient events critical for data valuation.

## 3. Preliminaries

### 3.1. Optimal Transport

OT provides a framework for comparing distributions by quantifying the minimal cost to transform a source $\mu_s = \sum_{i=1}^{n} a_i \delta_{z_i}$ into a target $\mu_t = \sum_{j=1}^{m} b_j \delta_{z'_j}$ (Villani et al., 2008; Peyré & Cuturi, 2019). Given a cost matrix $\mathbf{C}$ encoding pairwise distances, the Wasserstein distance finds an optimal transport plan $\mathbf{T}$ where each entry $T_{ij} \geq 0$ specifies the mass moved from $z_i$ to $z'_j$:

$$\mathcal{W}(\mu_s, \mu_t) = \min_{\mathbf{T} \in \Pi(\mu_s, \mu_t)} \langle \mathbf{T}, \mathbf{C} \rangle, \quad (1)$$

where $\Pi(\mu_s, \mu_t)$ enforces mass conservation: $\mathbf{T}\mathbf{1}_m = \mathbf{a}$ and $\mathbf{T}^\top \mathbf{1}_n = \mathbf{b}$.

**Unbalanced Optimal Transport (UOT).** Standard OT requires all mass to be transported, UOT relaxes this constraint through KL-divergence penalties (Chizat et al., 2018; Séjourné et al., 2019):

$$\mathcal{W}_{\text{UOT}}^{\kappa}(\mu_s, \mu_t) = \min_{\mathbf{T} \geq 0} \langle \mathbf{T}, \mathbf{C} \rangle + \kappa D_{\text{KL}}(\mathbf{T}\mathbf{1}_m \| \mathbf{a}) + \kappa D_{\text{KL}}(\mathbf{T}^\top \mathbf{1}_n \| \mathbf{b}), \quad (2)$$

where $\kappa > 0$ controls the trade-off between transport cost and mass conservation. This enables *selective matching*: dissimilar points can remain partially unmatched rather than forced into high-cost alignments. The dual formulation yields optimal potentials $(\mathbf{f}^*, \mathbf{g}^*)$ satisfying $\nabla_{\mathbf{a}} \mathcal{W}_{\text{UOT}}^{\kappa} = \mathbf{f}^*$ and $\nabla_{\mathbf{b}} \mathcal{W}_{\text{UOT}}^{\kappa} = \mathbf{g}^*$, which encode each point's marginal contribution to the discrepancy.

### 3.2. Problem Formulation

**Setup.** Let $\mathbf{X}_{\text{eval}} \in \mathbb{R}^{T_{\text{eval}} \times d}$ denote a time series to be valued, and $\mathbf{X}_{\text{ref}} \in \mathbb{R}^{T_{\text{ref}} \times d}$ a reference time series representing the

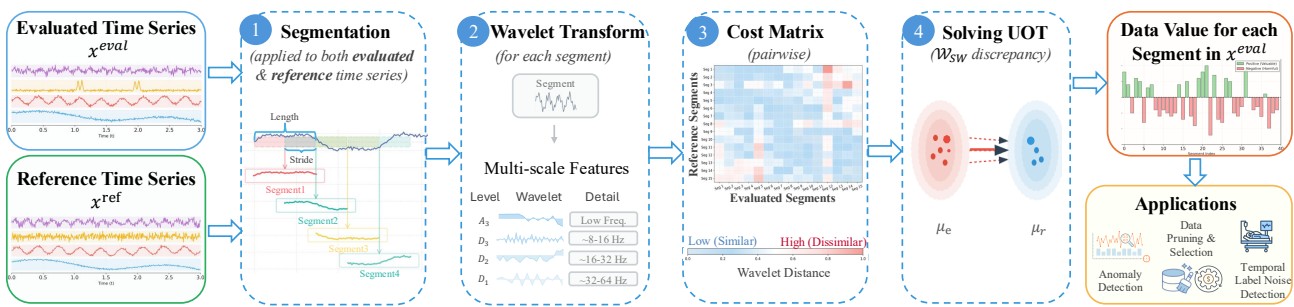

*Figure 1.* **Overview of the TIMELAVA Framework.** Given an evaluated time series and a reference time series, TIMELAVA assigns a value to each temporal segment indicating its quality relative to the reference. Both series are partitioned into overlapping segments and transformed into multi-scale wavelet representations. The Selective Wavelet-based Wasserstein ($\mathcal{W}_{\text{SW}}$) discrepancy is then computed between the two segment distributions. Segment values are efficiently derived from the optimal dual potentials without model training, enabling diverse applications including anomaly detection, data pruning, and label noise detection.

target distribution against which data quality is assessed. The specific choice of reference depends on the application, for example, validation sequences for forecasting tasks, normal segments for anomaly detection, or verified labeled data for noisy label detection. To preserve temporal dependencies, we partition each time series into overlapping segments using a sliding window of length $L$ with stride $S$, yielding $n$ evaluated segments $\mathcal{D}_{\text{eval}} = \{(\mathbf{x}_i, y_i)\}_{i=1}^n$ and $m$ reference segments $\mathcal{D}_{\text{ref}} = \{(\mathbf{x}'_j, y'_j)\}_{j=1}^m$, where $\mathbf{x}_i, \mathbf{x}'_j \in \mathbb{R}^{L \times d}$ and labels $y_i, y'_j \in \mathcal{Y}$ encode task-specific information when available. These segments induce empirical distributions $\mu_{\text{eval}}$ and $\mu_{\text{ref}}$ over the segment space $\mathcal{X}$.

**Objective.** We aim to construct a valuation function $v: \mathcal{X} \to \mathbb{R}$ that quantifies each segment's intrinsic contribution to downstream task performance without pre-specifying a learning algorithm. Formally, the valuation function must satisfy the following properties:

(1) *Learning-agnostic*: Independent of any specific model architecture or training procedure, generalizing across diverse tasks;
(2) *Robustness*: Stable under non-stationarity and distributional shifts inherent to real-world time series;
(3) *Computational efficiency*: Scalable to large datasets without requiring model retraining for each data point.

## 4. The TIMELAVA Framework

We now present the TIMELAVA framework for learning-agnostic time series data valuation. Our approach addresses two fundamental challenges: (1) defining a meaningful distance between temporal segments that captures both transient events and sustained patterns, and (2) computing robust valuations under the non-stationarity inherent to real-world time series. To this end, we adopt a wavelet-based segment distance that captures multi-scale temporal structure (§4.1), and integrate it with UOT to enable robust selective match-

ing (§4.2). We then derive an efficient valuation scheme via sensitivity analysis of the dual formulation (§4.3). Figure 1 illustrates the complete workflow.

### 4.1. Capturing Temporal Patterns with Wavelets

A central objective in time series data valuation is to identify the segments that contribute most positively or negatively to downstream task performance. These influential segments can manifest in multiple forms: persistent trends shaping long-term behavior, localized dynamics capturing short-term fluctuations, or transient events.

**Why Wavelets?** Fourier transforms decompose signals into global frequency components but discard temporal locality (Bloomfield, 2004). The Discrete Wavelet Transform (DWT) addresses this limitation by providing a joint time-frequency representation (Heil & Walnut, 1989; Rhif et al., 2019). As stated in Figure 2, while Fourier analysis reveals overall frequency content, wavelets preserve both the timing and frequency characteristics of transient events. This localization property is particularly valuable for non-stationary time series where distributional properties evolve over time.

**Definition 4.1** (Discrete Wavelet Transform (Heil & Walnut, 1989))**.** For a segment $\mathbf{x} = [x_0, x_1, \ldots, x_{L-1}]^\top \in \mathbb{R}^L$, the DWT coefficients are:

$$\Psi(\mathbf{x})_{j,k} = \sum_{l=0}^{L-1} x_l \psi_{j,k}[l], \quad j, k \in \mathbb{Z}, \tag{3}$$

where $\psi_{j,k}[l] = 2^{-j/2}\psi(2^{-j}l - k)$ are wavelets obtained by scaling and translating a mother wavelet $\psi$. For multivariate series, the DWT is applied independently along each feature dimension.

**Definition 4.2** (Wavelet Distance)**.** For two segments $\mathbf{x}_i, \mathbf{x}_j \in \mathbb{R}^L$, the wavelet distance is:

$$d_{\text{wav}}(\mathbf{x}_i, \mathbf{x}_j) = \|\Psi(\mathbf{x}_i) - \Psi(\mathbf{x}_j)\|_1. \tag{4}$$

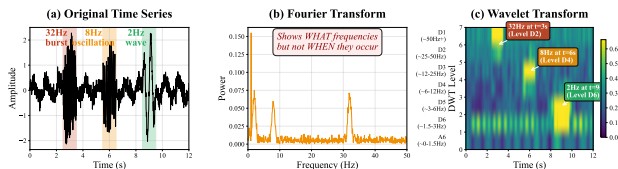

*Figure 2.* **Wavelet vs. Fourier Transform.** (a) A signal containing a low-frequency baseline and two localized high-frequency events. (b) Fourier transform captures overall frequency content but loses temporal localization. (c) Wavelet transform preserves both time and frequency, enabling precise localization of transient events.

**Implementation.** We use the Daubechies-4 (db4) wavelet with 2-level decomposition in all experiments, which balances temporal localization and frequency resolution.

**Lemma 4.3** (Proof in Appendix B.1). *The wavelet distance $d_{\text{wav}}$ is a valid metric, satisfying non-negativity, identity of indiscernibles, symmetry, and the triangle inequality.*

### 4.2. Selective Wavelet-based Wasserstein Discrepancy

Having defined a distance between segments, we now formalize the valuation principle and address the challenges posed by non-stationary time series.

**Why UOT?** Real-world time series exhibit *non-stationarity*: statistical properties evolve over time, creating coexisting temporal modes. Standard OT enforces strict mass conservation (Eq. 1), forcing every segment to be matched even across fundamentally different modes or to isolated outliers. This constraint causes two problems: (1) erroneous pairings that inflate discrepancy estimates, and (2) unbounded sensitivity to outliers, where a single anomaly can dominate the distance (Fatras et al., 2021).

To overcome this vulnerability, we adopt UOT (Eq. 2), which relaxes mass conservation through KL divergence penalties. As illustrated in Figure 3, this enables *selective matching*: dissimilar segments can remain partially unmatched rather than forced into high-cost alignments. This mechanism addresses both modal shifts from non-stationarity and isolated outliers from sensor noise or data corruption, both of which manifest computationally as high-cost mismatches that destabilize standard OT.

Building on these principles, we define the Selective Wavelet-based Wasserstein discrepancy, integrating the robustness of UOT with the wavelet-based ground metric.

**Definition 4.4** (Selective Wavelet-based Wasserstein Discrepancy). Let $\mu_{\text{eval}}$ and $\mu_{\text{ref}}$ denote the evaluated and reference segment distributions. The Selective Wavelet-based Wasserstein discrepancy is:

$$\mathcal{W}_{\text{SW}}(\mu_{\text{eval}}, \mu_{\text{ref}}) := \min_{\mathbf{T} \geq 0} \langle \mathbf{T}, \mathbf{D} \rangle + \kappa D_{\text{KL}}(\mathbf{T}\mathbf{1}_m \| \boldsymbol{\Delta}_n)$$
$$+ \kappa D_{\text{KL}}(\mathbf{T}^\top \mathbf{1}_n \| \boldsymbol{\Delta}_m), \qquad (5)$$

where $\kappa > 0$ controls marginal relaxation, $\boldsymbol{\Delta}_n = \mathbf{1}_n/n$ and $\boldsymbol{\Delta}_m = \mathbf{1}_m/m$, and the cost matrix is:

$$\mathbf{D}_{ij} = \underbrace{d_{\text{wav}}(\mathbf{x}_i, \mathbf{x}_j')}_{\text{wavelet distance}} + \underbrace{c \cdot \mathcal{W}_{d_{\text{wav}}}(\mu_{\text{eval}}(\cdot|y_i), \mu_{\text{ref}}(\cdot|y_j'))}_{\text{label consistency}} \quad (6)$$

Here $d_{\text{wav}}$ is the wavelet distance (Eq. 4), $\mu(\cdot \mid y)$ is the conditional distribution given label $y$. $c \geq 0$ balances feature similarity and label consistency.

### 4.3. Efficient Valuation via Dual Sensitivity Analysis

Having defined the Selective $\mathcal{W}_{\text{SW}}$ discrepancy, we now derive how to efficiently compute the value of each segment without model retraining.

**Valuation via Distributional Alignment.** Following the learning-agnostic paradigm of LAVA (Just et al., 2023), we define a segment's value through its contribution to distributional alignment: segments that help reduce the discrepancy between $\mu_{\text{eval}}$ and $\mu_{\text{ref}}$ are beneficial, while those that increase it are detrimental. The key insight is that OT-based distributional discrepancy is theoretically linked to generalization (formalized in Section 5), justifying using $\mathcal{W}_{\text{SW}}$ to assess data quality without training any predictive model.

**Segment Valuation.** The dual formulation of UOT provides an elegant solution: dual potentials represent the marginal cost of transporting mass at each location (Villani et al., 2008). A segment with high dual potential is costly to match to the reference, indicating it deviates from $\mu_{\text{ref}}$, while a segment with low potential matches easily, indicating alignment.

**Theorem 4.5** (Segment Valuation (Proof in Appendix B.4)). *Let $(\mathbf{f}^*, \mathbf{g}^*)$ be the optimal dual variables from solving the $\mathcal{W}_{\text{SW}}$ problem (Eq. 5), and define $\psi_\kappa(u) = \kappa(1 - e^{-u/\kappa})$. The value of evaluated segment $(\mathbf{x}_i, y_i)$ is:*

$$v(\mathbf{x}_i) = -\left(\psi_\kappa(\mathbf{f}_i^*) - \frac{1}{n-1}\sum_{j \neq i} \psi_\kappa(\mathbf{f}_j^*)\right). \qquad (7)$$

The negative sign in Eq. 7 ensures that segments with low dual potentials receive positive values, while the centering term provides relative assessment against the average contribution of other segments. Thus, $v(\mathbf{x}_i) > 0$ indicates the segment helps align the distributions, while $v(\mathbf{x}_i) < 0$ indicates it hinders alignment. In practice, we use entropy regularization (Cuturi, 2013; Séjourné et al., 2019) for efficient computation. Theorem B.2 (Appendix B.5) shows that as the regularization strength $\varepsilon \to 0$, the approximate values converge to the true values with ranking preserved. Algorithm 1 summarizes the complete procedure.

**Point-wise Valuation.** While segment-level values are useful for coarse-grained data curation, many applications require finer temporal resolution. For instance, in anomaly

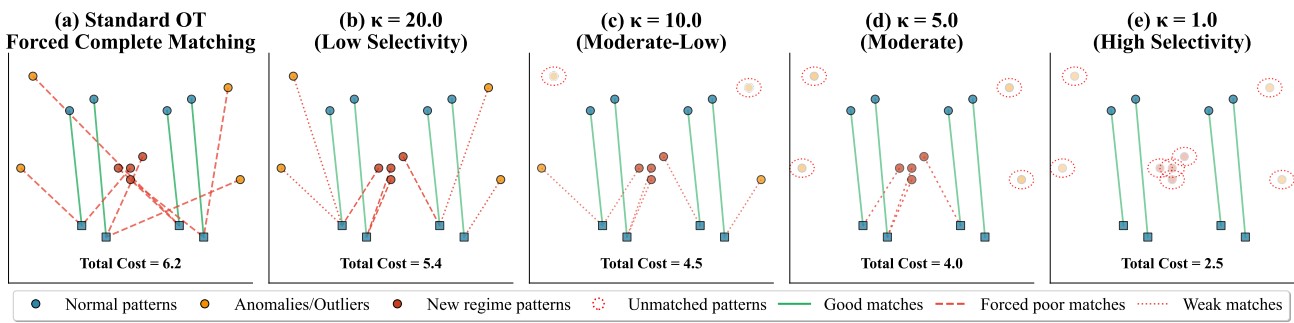

*Figure 3.* **UOT vs. OT.** (a) Standard OT forces complete matching, creating costly alignments for outliers and regime shifts. (b–e) As $\kappa$ decreases, UOT becomes more selective, allowing dissimilar points to remain unmatched rather than forced into poor alignments.

detection, we need to pinpoint the exact time points where anomalies occur rather than entire segments. Since our sliding window approach produces overlapping segments, each time point appears in multiple segments. We aggregate these segment-level values to obtain point-wise scores:

$$v(t) = \frac{1}{|\mathcal{S}_t|} \sum_{\mathbf{x}_i \in \mathcal{S}_t} v(\mathbf{x}_i), \qquad (8)$$

where $\mathcal{S}_t$ is the set of segments containing time point $t$. This aggregation provides context-aware estimation by considering each point's contribution across temporal contexts.

**Computational Complexity.** The computational complexity is dominated by three components: wavelet transforms require $O((n + m)L)$ using the fast pyramidal algorithm, pairwise cost matrix computation requires $O(nmL)$, and entropy-regularized UOT solving requires $O(nm \log(1/\varepsilon))$. The overall complexity is $O(nmL + nm \log(1/\varepsilon))$.

## 5. Theoretical Analysis

Having established the TIMELAVA framework, we now provide theoretical guarantees justifying its design. We prove that $\mathcal{W}_{\text{SW}}$ (1) exhibits bounded sensitivity to contamination, enabling the distinction between isolated anomalies and regime shifts (§5.1), and (2) justifies its use as a valuation proxy by linking distributional discrepancy to model-agnostic generalization (§5.2). These theoretical guarantees distinguish our approach from heuristic valuation methods and justify the learning-agnostic paradigm for time series.

**Notation.** For theoretical analysis involving labels, we denote the joint distributions over segments and labels as $\mu_e^{f_e}$ and $\mu_r^{f_r}$, where $f_e : \mathcal{X} \to \mathcal{Y}$ and $f_r : \mathcal{X} \to \mathcal{Y}$ are the labeling functions for evaluated and reference data. This generalizes the notation $\mu_{\text{eval}}$ and $\mu_{\text{ref}}$ from Section 3 to explicitly account for potentially different label distributions.

### 5.1. Robustness to Non-stationarity

We now formalize the robustness arguments from §4.2 by proving that $\mathcal{W}_{\text{SW}}$ exhibits bounded sensitivity to contami-

nation. Unlike standard OT where costs grow unboundedly with outlier distance (Fatras et al., 2021), UOT ensures stability under non-stationarity.

**Theorem 5.1** (Robustness to Outlier Contamination (Proof in Appendix B.2)). *Consider an evaluated distribution $\mu_e^{f_e}$ contaminated by an outlier mode $(\mathbf{z}, y_z)$, resulting in $\tilde{\mu}_e^{f_e} = \zeta \delta_{(\mathbf{z}, y_z)} + (1 - \zeta) \mu_e^{f_e}$ with relative mass $\zeta \in (0, 1)$. The $\mathcal{W}_{\text{SW}}$ discrepancy between the contaminated and reference distributions satisfies:*

$$\begin{aligned} \mathcal{W}_{\text{SW}}(\tilde{\mu}_e^{f_e}, \mu_r^{f_r}) &\le (1 - \zeta) \mathcal{W}_{\text{SW}}(\mu_e^{f_e}, \mu_r^{f_r}) \\ &\quad + 2\zeta\kappa \left(1 - e^{-\bar{d}(\mathbf{z}, y_z)/(2\kappa)}\right), \end{aligned} \qquad (9)$$

*where $\bar{d}(\mathbf{z}, y_z)$ is the average cost $\mathbf{D}$ of transporting the outlier to samples in $\mu_r^{f_r}$.*

This result shows that even an infinitely distant outlier ($\bar{d} \to \infty$) contributes at most $2\zeta\kappa$ to the discrepancy, in stark contrast to standard OT where the impact is unbounded (Fatras et al., 2021). Since non-stationarity manifests as collections of mismatched points, this single-outlier stability naturally extends to regime shifts, with total influence bounded proportionally to the new regime's mass. This bounded-influence property makes the resulting valuations both stable and informative: isolated anomalies produce extreme negative scores, while segments within a regime shift receive moderately low scores. The extremity of the score thus distinguishes genuine anomalies from systemic shifts (see Appendix B.2).

### 5.2. Connection to Learning-Agnostic Generalization

To justify using distributional alignment for data valuation, we prove that $\mathcal{W}_{\text{SW}}$ directly controls the gap between training and reference performance. We consider any predictive model $f : \mathcal{X} \to [0, 1]$ with loss function $\mathcal{L} : \mathcal{Y} \times [0, 1] \to \mathbb{R}_+$ that is $k$-Lipschitz in both arguments. We assume $f$ is $\epsilon$-Lipschitz with respect to $d_{\text{wav}}$, and that predictions and labels are bounded: $|f(\mathbf{x})|, |y| \le M$. We first introduce a regularity condition on labeling functions.

**Definition 5.2** (Probabilistic Cross-Lipschitzness in the Wavelet Domain)**.** Two labeling functions $f_e : \mathcal{X} \to \mathcal{Y}$ and $f_r : \mathcal{X} \to \mathcal{Y}$ are $(\epsilon_{er}, \delta_{\text{wav}})$-probabilistic cross-Lipschitz with respect to a coupling $\pi$ over $\mathcal{X} \times \mathcal{X}$ if:

$$\mathbb{P}_{(\mathbf{x}_e, \mathbf{x}_r) \sim \pi} \left[ |f_e(\mathbf{x}_e) - f_r(\mathbf{x}_r)| > \epsilon_{er} \cdot d_{\text{wav}}(\mathbf{x}_e, \mathbf{x}_r) \right] \leq \delta_{\text{wav}}. \tag{10}$$

Intuitively, this condition requires that the two labeling functions assign similar labels to time series segments that are close in the wavelet domain with high probability.

**Theorem 5.3** (Performance Bound (Proof in Appendix B.3))**.** *Under the regularity conditions above, if $f_e$ and $f_r$ are $(\epsilon_{er}, \delta_{\text{wav}})$-probabilistic cross-Lipschitz with respect to the optimal coupling from $\mathcal{W}_{\text{SW}}$, then:*

$$\mathbb{E}_{\mu_r^{f_r}}[\mathcal{L}(y, f(\mathbf{x}))] \leq \mathbb{E}_{\mu_e^{f_e}}[\mathcal{L}(y, f(\mathbf{x}))]$$
$$+ k\epsilon \, \mathcal{W}_{\text{SW}}(\mu_e^{f_e}, \mu_r^{f_r}) + 2kM\delta_{\text{wav}}. \tag{11}$$

This bound establishes that $\mathcal{W}_{\text{SW}}$ controls model-agnostic generalization: smaller discrepancy guarantees better performance for any well-behaved model. This directly justifies our valuation scheme (Theorem 4.5): a segment's marginal contribution to reducing $\mathcal{W}_{\text{SW}}$, captured by its dual potential, quantifies its impact on downstream tasks without training any specific model. Compared to LAVA's i.i.d. bound (Just et al., 2023), our result accounts for temporal structure via wavelets and provides robustness via UOT. Combined with Theorem 5.1, this establishes TimeLAVA as both stable under non-stationarity and theoretically grounded for learning-agnostic valuation.

# 6. Use Cases of TimeLAVA

We conduct a comprehensive empirical study to evaluate TimeLAVA across diverse time series data valuation scenarios. Our experiments span multiple real-world datasets and tasks, assessing point-wise valuation in anomaly detection, and segment-wise valuation in both data selection/pruning and temporal label noise detection, thereby covering both predictive and diagnostic use cases.

## 6.1. Anomaly Detection

In many time series applications, parts of the training data may contain anomalies from sensor faults, rare events, or unexpected operational states that deviate from normal temporal dynamics (Xu et al., 2022). This task aims to evaluate whether TimeLAVA's value estimates can distinguish anomalous from normal segments without prior location information, enabling robust anomaly identification.

**Experimental Setup.** Following (Jiang et al., 2022; Zhang et al., 2025), all methods compute anomaly scores directly

on potentially contaminated data with unknown anomaly proportion. TimeLAVA approaches this as a data valuation problem: we compute point-wise values via Eq. 8 as anomaly scores, identifying low-value points as anomalies. This aligns with the intuition that anomalies increase distributional discrepancy with the reference set (Theorem 4.5). A clean validation series serves as the reference distribution, and we set $c = 0$ in Eq. 6 for unsupervised detection.

**Datasets.** We evaluate TimeLAVA on several widely used benchmark datasets: UCR (Wu & Keogh, 2021), NAB (Ahmad et al., 2017), SMD (Su et al., 2019), SMAP and MSL (Hundman et al., 2018), PSM (Abdulaal et al., 2021), SWaT (Mathur & Tippenhauer, 2016), WADI (Ahmed et al., 2017), and KDD-CUP99 (Stolfo et al., 2000). These datasets span diverse domains and exhibit varying sampling frequencies, sequence lengths, and anomaly characteristics, providing a comprehensive basis for evaluation.

**Baselines.** We compare TimeLAVA against two categories of baselines: (1) time series data valuation methods: LWCV (Ghosh et al., 2020), and the recent state-of-the-art TimeInf (Zhang et al., 2025), and (2) anomaly detection algorithms: Isolation Forest (Liu et al., 2008), and an LSTM-based detector (Hundman et al., 2018), recent deep learning approaches (DCdetector (Yang et al., 2023), Anomaly Transformer (Xu et al., 2022), ModernTCN (Luo & Wang, 2024) and CATCH (Wu et al., 2025)).

**Results.** We qualitatively evaluate on the UCR (Wu & Keogh, 2021), examining how different methods assign values to normal and anomalous across four representative anomaly types (Fig. 4). TimeLAVA produces discriminative scores that robustly identify diverse anomalies with high values, while keeping scores in normal regions low to prevent false alarms. Quantitative evaluations on multivariate benchmark datasets (Table 1) align with these qualitative findings. TimeLAVA attains the highest or near-highest AUC and F1 scores across all datasets, while preserving competitive computational efficiency. These results demonstrate that TimeLAVA's valuation scores accurately reflect true data quality: segments assigned negative values correspond to anomalies, validating the effectiveness of learning-agnostic valuation for time series quality assessment. Additional experimental details, ablation studies validating both UOT and wavelet components, and parameter sensitivity analyses are provided in Appendix C.2.

## 6.2. Data Pruning and Selection

A primary objective of data valuation is to identify high-quality segments that improve model performance while detecting corrupted or low-value segments that degrade it. This section evaluates TimeLAVA's ability to produce meaningful value rankings for time series segments.

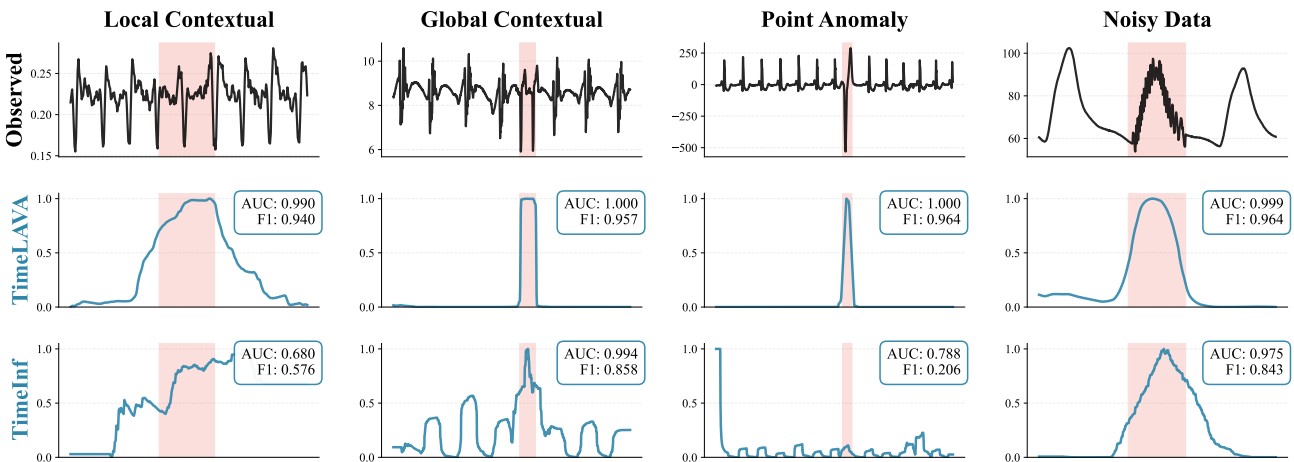

*Figure 4.* **Qualitative analysis on the UCR Time Series Anomaly Archive.** Each column corresponds to one anomaly type: *Local Contextual*, *Global Contextual*, *Point Anomaly*, and *Noisy Data*. The first row shows the observed time series with ground-truth anomalous regions shaded in red. Subsequent rows plot the normalized negated anomaly scores produced by each method. Boxes show AUC/F1 scores (higher is better); higher curves in red regions with flat responses elsewhere indicate better localization and fewer false alarms.

*Table 1.* **Quantitative evaluation of anomaly detection performance and runtime.** Higher AUC and F1 scores indicate better detection accuracy. Across four real-world datasets, TIMELAVA achieves superior or competitive accuracy over established baselines, while maintaining low computational cost.

| | SMD | | | SMAP | | | MSL | | | PSM | | |
|---|---|---|---|---|---|---|---|---|---|---|---|---|
| | AUC | F1 | Time (s) | AUC | F1 | Time (s) | AUC | F1 | Time (s) | AUC | F1 | Time (s) |
| Isolation Forest | 0.82 | 0.37 | 0.30 | 0.67 | 0.33 | 0.14 | 0.68 | 0.31 | 0.11 | 0.71 | 0.51 | 1.12 |
| LSTM | 0.79 | 0.30 | 307.08 | 0.55 | 0.31 | 245.52 | 0.70 | 0.34 | 8.77 | 0.62 | 0.38 | 817.13 |
| LWCV | 0.79 | 0.29 | 3369.52 | 0.63 | 0.26 | 354.30 | 0.65 | 0.32 | 109.84 | 0.56 | 0.34 | 1852.07 |
| DCdetector | 0.70 | 0.25 | 106.70 | 0.67 | 0.35 | 194.52 | 0.69 | 0.34 | 50.13 | 0.49 | 0.27 | 4075.59 |
| Anomaly Transformer | 0.51 | 0.05 | 246.80 | 0.49 | 0.04 | 51.24 | 0.42 | 0.10 | 20.13 | 0.48 | 0.15 | 2823.42 |
| ModernTCN | 0.72 | 0.15 | 225.66 | 0.46 | 0.12 | 650.70 | 0.63 | 0.12 | 81.56 | 0.59 | 0.09 | 4257.66 |
| CATCH | 0.81 | 0.24 | 3332.20 | 0.50 | 0.06 | 1314.04 | 0.66 | 0.14 | 677.21 | 0.65 | 0.12 | 1367.47 |
| TimeInf | 0.87 | 0.25 | 20.85 | 0.73 | 0.34 | 10.59 | 0.72 | 0.35 | 6.17 | 0.63 | 0.02 | 350.10 |
| **TIMELAVA (Ours)** | **0.91** | **0.52** | 14.01 | **0.74** | **0.54** | 0.72 | **0.81** | **0.49** | 0.25 | **0.77** | **0.58** | 8.16 |

**Experimental Setup.** We evaluate data valuation methods through two complementary experiments: (i) Data Selection: retaining only the top-$k$% highest-valued segments for training, and (ii) Data Pruning: progressively removing the lowest-valued segments. To enable controlled evaluation with known ground truth, we inject synthetic noise into 20% of randomly selected training segments, simulating four common corruption types: Gaussian noise, spike artifacts, linear drift, and scale perturbations. This semi-synthetic approach enables two types of evaluation: downstream forecasting performance after data selection/pruning, and corruption detection accuracy against known labels. All data valuation methods compute per-segment quality scores, rank segments accordingly, and either retain top-$k$% (selection) or remove bottom-$k$% (pruning). We then train a linear AR model on the selected segments and evaluate forecasting performance (RMSE, $R^2$) on a held-out test set. We set $c = 0$ in Eq. 6 since these forecasting tasks focus on temporal pattern valuation without label information.

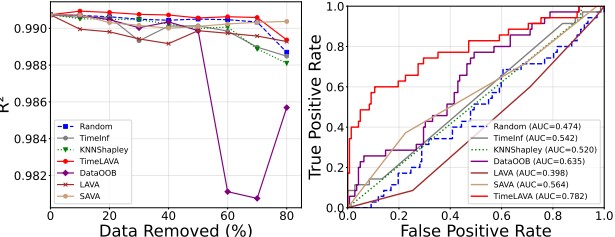

*Figure 5.* Data pruning performance ($R^2$) and noise detection (ROC) on Exchange dataset with 20% corrupted segments.

**Datasets.** We evaluate on four benchmark time series datasets commonly used in forecasting literature: ETTh1 (Zhou et al., 2021), Traffic, Exchange, and Electricity (Lai et al., 2018). Each dataset is segmented into fixed-length, non-overlapping segments and divided into training, validation, and test sets, with the validation set serving as the reference time series.

**Baselines.** We evaluate TIMELAVA against several representative data valuation methods: TimeInf (Zhang et al., 2025) adapts influence function to time series by estimating the effect of removing a segment on test loss. KNN-Shapley (Jia et al., 2019a) approximates Shapley values via $k$NN-based marginal utility estimation. Data-OOB (Kwon & Zou, 2023) assigns values using out-of-bag predictions from a bagged decision-tree ensemble. LAVA (Just et al., 2023) measures training–validation alignment via the Wasserstein distance, and SAVA (Kessler et al., 2025) provides a scalable batch-based variant. A Random baseline is also included for reference.

**Results.** Fig. 5 demonstrates TIMELAVA's effective performance in both data quality assessment and noise detection on the Exchange dataset. In the data pruning experiment (left), TIMELAVA maintains the highest $R^2$ as low-value segments are removed, with performance degrading only after removing 45% of the data, indicating accurate identification of detrimental segments. Other methods show earlier performance degradation, with Data-OOB experiencing a sharp drop after 50% removal. The ROC curve (right) evaluates corruption detection: since we know which segments were corrupted during injection, we can measure each method's ability to identify them via standard binary classification metrics. TIMELAVA achieves the highest AUC (0.782) for detecting corrupted segments, substantially outperforming all baselines including TimeInf (0.542) and KNN-Shapley (0.520). Additional experimental details and results across all datasets are provided in Appendix C.3.

### 6.3. Noisy label detection

In real-world detection systems, label quality often varies over time due to operational conditions, annotator fatigue, or system degradation (Carpenter, 2003; Hsieh & Kocielnik, 2016; Nagaraj et al., 2025). Traditional approaches assume static noise distributions, fundamentally misspecifying the noise model and leading to suboptimal performance. This task evaluates whether TIMELAVA's temporal modeling can effectively identify different time-varying label corruption patterns.

**Experimental Setup.** Following established practices (Just et al., 2023; Jiang et al., 2023), we adopt a semi-synthetic framework that treats original dataset labels as ground truth and systematically injects temporal noise patterns. Labels are corrupted by flipping to the opposite class at segments selected according to four temporal corruption patterns with average noise rate $\eta \in \{0.05, 0.10, 0.15, 0.20\}$: *Random*, *Periodic*, *Decay* and *Growth* (Nagaraj et al., 2025). We split each dataset temporally into 70% evaluated data (with injected noise) and 30% reference data (clean). All methods compute segment-level value scores and rank segments to identify those most likely to be mislabeled. We evaluate

detection performance using F1 score. We set $c = 1$ in Eq. 6 to incorporate label information. We compare against the same baseline methods used in §6.2.

**Datasets.** We evaluate on three healthcare classification datasets: Moving (Reyes-Ortiz et al., 2013), Senior (Logacjov & Ustad, 2023), and Blinking (Roesler, 2013). These tasks involve sequential labeling where temporal corruption patterns can realistically occur in practical applications.

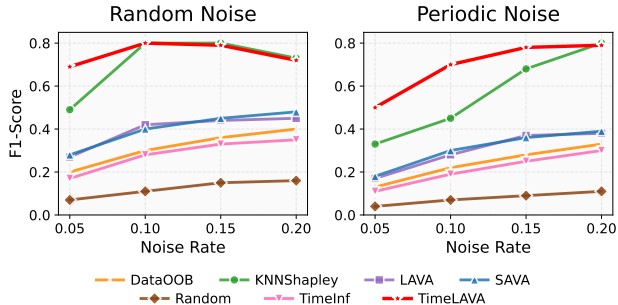

*Figure 6.* F1 scores for noisy label detection on Moving dataset.

**Results.** Fig. 6 shows results on the Moving dataset for two representative noise patterns. For periodic noise, TIMELAVA consistently outperforms other methods across all noise rates, achieving F1 scores of 0.78 at 15% noise compared to 0.65 for TimeInf and 0.52 for KNN-Shapley, showing robust performance in realistic low-corruption scenarios. This superiority stems from TIMELAVA's ability to model temporal dependencies, enabling it to identify systematic patterns in label corruption that occur in real-world scenarios such as periodic sensor failures or shift-based annotation quality variations. For random noise, TIMELAVA performs comparably to KNN-Shapley, which is expected when corruption lacks temporal structure. This validates that our method exploits temporal patterns when present without overfitting when absent. Results for all datasets and noise patterns are provided in Appendix C.4.

### 6.4. Parameter Sensitivity

We conduct comprehensive sensitivity analyses on key hyperparameters (details in Appendix C.2). TIMELAVA is robust to wavelet basis choice (AUC ranges from 0.87 to 0.91 across different wavelet types and decomposition levels on SMD), confirming that db4 with 2-level decomposition provides a strong default without dataset-specific tuning. For domain-specific applications, practitioners may consider: higher-order wavelets (db6, db8) for smooth signals, lower-order wavelets (Haar, db2) for discontinuous or step-like signals, shallower decomposition (1–2 levels) for high-frequency transient detection, and deeper decomposition (3–4 levels) for slowly-varying or low-frequency data. Segment length and stride show stable performance:

AUC saturates around length 100–150, and remains stable for stride $\in [1, 10]$ (Appendix Figures 11–12). For the label consistency weight $c$ in Eq. 6, performance follows an inverted-U shape peaking at $c = 1$ and remains stable for $c \in [0.5, 2.0]$ (Appendix Figure 25), as underweighting label information or over-emphasizing label alignment both degrade performance.

# 7. Conclusion

This paper introduced TIMELAVA, a learning-agnostic framework for valuing temporal segments in time series data. By combining multi-scale wavelet transforms with unbalanced optimal transport, the proposed $\mathcal{W}_{SW}$ discrepancy captures localized temporal patterns while remaining resilient to non-stationarity. Segment values are efficiently derived via dual sensitivity analysis without model training, with theoretical guarantees linking $\mathcal{W}_{SW}$ to model-agnostic generalization and bounded outlier sensitivity. Extensive experiments demonstrate that TIMELAVA's value scores enable substantial improvements across anomaly detection, data pruning, and noisy label detection, consistently outperforming existing valuation methods. These results validate that learning-agnostic valuation provides a principled approach to quantifying intrinsic data quality that translates directly into downstream task performance.

**Limitations and Future Work.** The effectiveness of TIMELAVA depends on the representativeness of the reference set: biased or incomplete references may undervalue rare yet important patterns. Mitigating this through diversity-aware sampling (e.g., $k$-medoids in wavelet feature space) or bootstrap averaging across multiple reference subsets is a promising direction. The current per-channel processing also does not explicitly model cross-channel dependencies; incorporating such modeling while preserving the learning-agnostic principle remains an open challenge. Additionally, adopting mini-batch strategies similar to SAVA (Kessler et al., 2025) would improve scalability to sequences with millions of time points. Finally, TIMELAVA's valuation scores naturally extend to tasks such as active learning and time series classification, where segment values can guide annotation priorities or identify the most informative training samples.

# Acknowledgements

This research was supported by the University of Melbourne's Research Computing Services and the Petascale Campus Initiative. WL was supported by a Melbourne Research Scholarship. EG and DS were supported by the Responsible AI Research Centre (RAIR). MG was supported by ARC grants DP240102088, and WIS-MBZUAI grant 142571.

# Impact Statement

This work contributes to the fields of machine learning and data-centric AI, with implications for time series applications in domains such as healthcare, finance, and industrial monitoring. The proposed TimeLAVA framework addresses a critical challenge in data curation: identifying which temporal segments are valuable for model training and which are detrimental due to noise, corruption, or distributional shift. By providing learning-agnostic valuation scores, our method can reduce the computational and financial costs of training on low-quality data, accelerate model development cycles, and improve deployed system reliability. In safety-critical domains such as medical diagnosis or financial risk assessment, more accurate identification of data quality issues could reduce the risk of models learning from corrupted or biased data. However, practitioners should be aware of important limitations. The valuation quality depends critically on the representativeness of the reference distribution: a biased or incomplete reference set may cause rare but important patterns to be undervalued. In domains with evolving data distributions, reference sets must be periodically updated to remain valid.

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

# Appendix

## Table of Contents

# A. Notation Summary

*Table 2.* Table of Notations

| Symbol | Description |
|---|---|
| **Time Series and Data Structures** | |
| $X_{\text{eval}} \in \mathbb{R}^{T_{\text{eval}} \times d}$ | Evaluated time series |
| $X_{\text{ref}} \in \mathbb{R}^{T_{\text{ref}} \times d}$ | Reference (validation) time series |
| $\mathbf{x}_i, \mathbf{x}'_j \in \mathbb{R}^{L \times d}$ | Evaluated and reference time series segments of length $L$ |
| $t$ | Time index |
| $T$ | Total length of time series |
| $L$ | Length of each segment (window size) |
| $S$ | Stride for sliding window |
| $d$ | Number of features in multivariate time series |
| $\mathcal{D}_{\text{eval}}$ | Evaluated segment-label pairs |
| $\mathcal{D}_{\text{ref}}$ | Reference segment-label pairs |
| $n, m$ | Number of segments in $\mathcal{D}_{\text{eval}}$ and $\mathcal{D}_{\text{ref}}$ |
| $\mathcal{X}$ | Input space of time series segments |
| $\mathcal{Y}$ | Output or label space |
| $\mathcal{S}_t$ | Set of segments containing time point $t$ |
| **Functions and Models** | |
| $f : \mathcal{X} \to [0, 1]$ | Predictive model |
| $f_e, f_r$ | Labeling functions for evaluated and reference data |
| $L : \mathcal{Y} \times [0, 1] \to \mathbb{R}_+$ | Loss function |
| **Distributions and Measures** | |
| $\mu_{\text{eval}}, \mu_{\text{ref}}$ | Empirical distributions over evaluated and reference segments |
| $\mu_{\text{eval}}^{f_e}, \mu_{\text{ref}}^{f_r}$ | Joint distributions over segments and labels |
| $\mu_{\text{eval}}(\cdot|y), \mu_{\text{ref}}(\cdot|y)$ | Conditional distributions given label $y$ |
| $\delta_z$ | Dirac delta measure centered at $z$ |
| $\tilde{\mu}_e^{f_e}$ | Contaminated distribution |
| **Wavelet Transform** | |
| $\psi$ | Mother wavelet function |
| $\psi_{j,k}[t]$ | Discrete wavelet at scale $j$ and position $k$ |
| $\Psi(\cdot)$ | Discrete Wavelet Transform (DWT) operator |
| $\Psi(\mathbf{x})_{j,k}$ | DWT coefficient at scale $j$ and position $k$ |
| $d_{\text{wav}}(\mathbf{x}_i, \mathbf{x}_j)$ | Wavelet distance: $\|\Psi(\mathbf{x}_i) - \Psi(\mathbf{x}_j)\|_1$ |
| **Optimal Transport (OT)** | |
| $\mathbf{T} \in \mathbb{R}_+^{n \times m}$ | Transport plan (coupling matrix) |
| $\mathbf{C}$ | Cost matrix for standard OT |
| $\mathbf{D}$ | Joint feature-label cost matrix for $\mathcal{W}_{\text{SW}}$ distance |
| $\Pi(\mu_{\text{eval}}, \mu_{\text{ref}})$ | Set of valid transport plans |
| $\mathcal{W}(\mu_{\text{eval}}, \mu_{\text{ref}})$ | Standard Wasserstein distance |
| $\mathcal{W}_{d_{\text{wav}}}$ | Wasserstein distance with wavelet ground metric |
| $\mathcal{W}_{\text{SW}}$ | Selective Wavelet-based Wasserstein discrepancy |
| **Unbalanced OT and Regularization** | |
| $\kappa > 0$ | Regularization parameter for UOT marginal relaxation |
| $D_{\text{KL}}(\cdot\|\cdot)$ | Kullback-Leibler divergence |
| $\mathbf{\Delta}_n = \mathbf{1}_n / n$ | Uniform distribution over $n$ samples |

| Symbol | Description |
|---|---|
| $\boldsymbol{\Delta}_m = \mathbf{1}_m/m$ | Uniform distribution over $m$ samples |
| $\mathbf{f}^* \in \mathbb{R}^n, \mathbf{g}^* \in \mathbb{R}^m$ | Optimal dual potentials |
| $\mathbf{f}_\varepsilon^*, \mathbf{g}_\varepsilon^*$ | Optimal dual potentials with entropy regularization |
| $\psi_\kappa(u)$ | Transform function from KL penalty |
| $\varepsilon > 0$ | Entropy regularization strength |
| $H(\mathbf{T})$ | Entropy of transport plan |

**Theoretical Constants and Bounds**

| | |
|---|---|
| $k$ | Lipschitz constant of loss function $L$ |
| $\epsilon$ | Lipschitz constant of model $f$ w.r.t. $d_{\text{wav}}$ |
| $M$ | Upper bound on $\|f(\mathbf{x})\|$ and $\|y\|$ |
| $\epsilon_{er}$ | Probabilistic cross-Lipschitz constant |
| $\delta_{\text{wav}} \in [0,1]$ | Cross-Lipschitz violation probability |
| $\pi^*$ | Optimal coupling for $\mathcal{W}_{\text{SW}}$ |
| $\zeta \in (0,1)$ | Relative mass of outlier mode |
| $\bar{d}(z, y_z)$ | Average cost of transporting outlier |

**Data Valuation**

| | |
|---|---|
| $v(\mathbf{x}_i)$ | Data value of segment $\mathbf{x}_i$ |
| $v_\varepsilon(\mathbf{x}_i)$ | Entropy-regularized data value |
| $v(t)$ | Point-wise data value at time $t$ |
| $w(t, \mathbf{z}^{[m]})$ | Weight for point-wise aggregation |

**Miscellaneous**

| | |
|---|---|
| $\mathbf{1}_k \in \mathbb{R}^k$ | Column vector of ones |
| $\langle \cdot, \cdot \rangle$ | Frobenius inner product |
| $\|\cdot\|_1$ | $L_1$ norm (sum of absolute values) |
| $\|\cdot\|_\infty$ | $L_\infty$ norm (maximum absolute value) |
| $c \geq 0$ | Hierarchical weight in $\mathbf{D}^{(W)}$ |
| $\mathcal{N}$ | Set of corrupted time points (noise injection) |
| $\eta$ | Noise rate |

# B. Proofs

This section provides the detailed proofs for the lemmas and theorems presented in the main text.

## B.1. Proof of Lemma 4.3: Wavelet Distance Properties

We need to prove that the wavelet distance $d_{\text{wav}}(\mathbf{x}_i, \mathbf{x}_j) = \|\mathcal{W}(\mathbf{x}_i) - \mathcal{W}(\mathbf{x}_j)\|_1$ is a metric, satisfying the four metric properties: non-negativity, identity of indiscernibles, symmetry, and the triangle inequality.

*Proof.* Let $\mathbf{x}_i, \mathbf{x}_j, \mathbf{x}_k \in \mathbb{R}^T$ be time series segments (for simplicity, consider $d = 1$; the extension to $d > 1$ by applying DWT channel-wise and summing L1 norms maintains metric properties if the sum is used, or if applied to concatenated wavelet coefficients). Let $\mathcal{W}(\mathbf{x})$ denote the vector of wavelet coefficients for $\mathbf{x}$.

1. **Non-negativity:** For any time series $\mathbf{x}_i, \mathbf{x}_j$,

$$d_{\text{wav}}(\mathbf{x}_i, \mathbf{x}_j) = \|\mathcal{W}(\mathbf{x}_i) - \mathcal{W}(\mathbf{x}_j)\|_1 \geq 0$$

This follows directly from the definition of the L1-norm, which is always non-negative.

2. **Identity of indiscernibles:** We need to show $d_{\text{wav}}(\mathbf{x}_i, \mathbf{x}_j) = 0 \iff \mathbf{x}_i = \mathbf{x}_j$.

- ($\Rightarrow$): If $d_{\text{wav}}(\mathbf{x}_i, \mathbf{x}_j) = 0$, then $\|\mathcal{W}(\mathbf{x}_i) - \mathcal{W}(\mathbf{x}_j)\|_1 = 0$. By properties of norms, this implies $\mathcal{W}(\mathbf{x}_i) - \mathcal{W}(\mathbf{x}_j) = \mathbf{0}$, so $\mathcal{W}(\mathbf{x}_i) = \mathcal{W}(\mathbf{x}_j)$. Since the DWT (as typically implemented with a complete basis) is invertible, we can apply the inverse DWT $\mathcal{W}^{-1}$ to both sides:

$$\mathcal{W}^{-1}(\mathcal{W}(\mathbf{x}_i)) = \mathcal{W}^{-1}(\mathcal{W}(\mathbf{x}_j)) \implies \mathbf{x}_i = \mathbf{x}_j$$

- ($\Leftarrow$): If $\mathbf{x}_i = \mathbf{x}_j$, then $\mathcal{W}(\mathbf{x}_i) = \mathcal{W}(\mathbf{x}_j)$ since the DWT is a deterministic linear transform. Therefore, $\|\mathcal{W}(\mathbf{x}_i) - \mathcal{W}(\mathbf{x}_j)\|_1 = \|\mathbf{0}\|_1 = 0$.

3. **Symmetry:** We need to show $d_{\text{wav}}(\mathbf{x}_i, \mathbf{x}_j) = d_{\text{wav}}(\mathbf{x}_j, \mathbf{x}_i)$.

$$\begin{aligned}
d_{\text{wav}}(\mathbf{x}_i, \mathbf{x}_j) &= \|\mathcal{W}(\mathbf{x}_i) - \mathcal{W}(\mathbf{x}_j)\|_1 \\
&= \| - (\mathcal{W}(\mathbf{x}_j) - \mathcal{W}(\mathbf{x}_i))\|_1 \\
&= \|\mathcal{W}(\mathbf{x}_j) - \mathcal{W}(\mathbf{x}_i)\|_1 \quad (\text{since } \| - \mathbf{v}\|_1 = \|\mathbf{v}\|_1) \\
&= d_{\text{wav}}(\mathbf{x}_j, \mathbf{x}_i)
\end{aligned}$$

Thus, symmetry holds.

4. **Triangle inequality:** We need to show $d_{\text{wav}}(\mathbf{x}_i, \mathbf{x}_k) \le d_{\text{wav}}(\mathbf{x}_i, \mathbf{x}_j) + d_{\text{wav}}(\mathbf{x}_j, \mathbf{x}_k)$.

$$\begin{aligned}
d_{\text{wav}}(\mathbf{x}_i, \mathbf{x}_k) &= \|\mathcal{W}(\mathbf{x}_i) - \mathcal{W}(\mathbf{x}_k)\|_1 \\
&= \|\mathcal{W}(\mathbf{x}_i) - \mathcal{W}(\mathbf{x}_j) + \mathcal{W}(\mathbf{x}_j) - \mathcal{W}(\mathbf{x}_k)\|_1 \\
&\le \|\mathcal{W}(\mathbf{x}_i) - \mathcal{W}(\mathbf{x}_j)\|_1 + \|\mathcal{W}(\mathbf{x}_j) - \mathcal{W}(\mathbf{x}_k)\|_1 \quad (\text{by triangle inequality of L1-norm}) \\
&= d_{\text{wav}}(\mathbf{x}_i, \mathbf{x}_j) + d_{\text{wav}}(\mathbf{x}_j, \mathbf{x}_k)
\end{aligned}$$

Therefore, the triangle inequality holds.

Having proved all four properties, we conclude that $d_{\text{wav}}(\mathbf{x}_i, \mathbf{x}_j)$ is a metric. $\square$

### B.2. Proof of Theorem 5.1: Robustness to Non-stationarity

**Lemma B.1.** *Suppose that $\tilde{\alpha} = \zeta\delta_z + (1 - \zeta)\alpha$ is a distribution perturbed by a Dirac mode at $z$ with relative mass $\zeta \in (0, 1)$. For a sample $y^*$ in the support of $\beta$, (Fatras et al., 2021) demonstrates:*

$$\mathcal{W}(\tilde{\alpha}, \beta) \ge (1 - \zeta)\mathcal{W}(\alpha, \beta) + \zeta\left(D(z, y^*) - g(y^*) + \int g \, d\beta\right),$$

*where $D(z, y^*)$ is the deviation of $\delta_z$, and $g$ is the optimal dual potential of $\mathcal{W}(\alpha, \beta)$.*

*Proof.* This theorem builds upon the foundation of Lemma B.1 by (Fatras et al., 2021). To establish this robustness bound, we construct a parametric family of feasible transport plans and optimize over the parameter to obtain the tightest upper bound. Let $\mathbf{T}^*$ be the optimal transport plan for the clean problem $\tilde{\mathcal{W}}_{\text{SW}}(\mu_t^{f_t}, \mu_v^{f_v})$. We construct a parametric family of transport plans $\tilde{\mathbf{T}}_\phi$ with parameter $\phi \in [0, 1]$ as follows:

$$\tilde{\mathbf{T}}_\phi = (1 - \zeta)\mathbf{T}^* + \zeta\phi(\delta_{\mathbf{z}} \otimes \mathbf{\Delta}_m) = \begin{pmatrix} (1 - \zeta)\mathbf{T}^* \\ \frac{\zeta\phi}{m}\mathbf{1}_{1 \times m} \end{pmatrix} \tag{12}$$

The parameter $\phi$ controls how the outlier mass is handled: when $\phi = 1$, all outlier mass $\zeta$ is transported to the validation set; when $\phi = 0$, the outlier mass is completely destroyed (incurring only mass penalty); and for $\phi \in (0, 1)$, we have partial transport with the remainder destroyed.

Since $\tilde{\mathbf{T}}_\phi$ is a feasible transport plan for the contaminated problem, we have the upper bound $\tilde{\mathcal{W}}_{\text{SW}}(\tilde{\mu}_t^{f_t}, \mu_v^{f_v}) \le \text{Cost}(\tilde{\mathbf{T}}_\phi)$, where the cost function is

$$\text{Cost}(\tilde{\mathbf{T}}_\phi) = \langle \mathbf{D}, \tilde{\mathbf{T}}_\phi \rangle + \kappa D_{\text{KL}}(\tilde{\mathbf{T}}_\phi \mathbf{1}_m \| \tilde{\mu}_t^{f_t}) + \kappa D_{\text{KL}}(\tilde{\mathbf{T}}_\phi^\top \mathbf{1}_{n+1} \| \mu_v^{f_v}) \tag{13}$$

We now analyze each component of this cost function to express it in terms of the parameter $\phi$.

For the transport cost component, we have

$$\langle \mathbf{D}, \tilde{\mathbf{T}}_\phi \rangle = \langle \mathbf{D}, (1-\zeta)\mathbf{T}^* + \zeta\phi(\delta_\mathbf{z} \otimes \boldsymbol{\Delta}_m) \rangle \tag{14}$$

$$= (1-\zeta)\langle \mathbf{D}, \mathbf{T}^* \rangle + \zeta\phi \cdot \bar{d}(\mathbf{z}, y_z) \tag{15}$$

where $\bar{d}(\mathbf{z}, y_z) = \frac{1}{m} \sum_{j=1}^m \mathbf{D}((\mathbf{z}, y_z), (\mathbf{x}_j^v, y_j^v))$ is the average cost of transporting the outlier to the validation set.

For the first KL divergence term on the training side, the marginal distribution is $\tilde{\mathbf{T}}_\phi \mathbf{1}_m = (1-\zeta)\mathbf{T}^* \mathbf{1}_m + \zeta\phi\delta_\mathbf{z}$. Using the joint convexity of the KL divergence, we obtain

$$D_{\mathrm{KL}}(\tilde{\mathbf{T}}_\phi \mathbf{1}_m \| \tilde{\mu}_t^{f_t}) \le (1-\zeta)D_{\mathrm{KL}}(\mathbf{T}^* \mathbf{1}_m \| \mu_t^{f_t}) + \zeta D_{\mathrm{KL}}(\phi\delta_\mathbf{z} \| \delta_\mathbf{z}) \tag{16}$$

where the key calculation shows that $D_{\mathrm{KL}}(\phi\delta_\mathbf{z} \| \delta_\mathbf{z}) = \phi \log \phi - \phi + 1$ for the generalized KL divergence.

Similarly, for the second KL divergence term on the validation side, the marginal is $\tilde{\mathbf{T}}_\phi^\top \mathbf{1}_{n+1} = (1-\zeta)(\mathbf{T}^*)^\top \mathbf{1}_n + \zeta\phi\boldsymbol{\Delta}_m$, which gives us

$$D_{\mathrm{KL}}(\tilde{\mathbf{T}}_\phi^\top \mathbf{1}_{n+1} \| \mu_v^{f_v}) \le (1-\zeta)D_{\mathrm{KL}}((\mathbf{T}^*)^\top \mathbf{1}_n \| \mu_v^{f_v}) + \zeta D_{\mathrm{KL}}(\phi\boldsymbol{\Delta}_m \| \boldsymbol{\Delta}_m) \tag{17}$$

where $D_{\mathrm{KL}}(\phi\boldsymbol{\Delta}_m \| \boldsymbol{\Delta}_m) = \phi \log \phi - \phi + 1$ by the same calculation.

Combining all cost components, we obtain

$$\mathrm{Cost}(\tilde{\mathbf{T}}_\phi) \le (1-\zeta)\tilde{\mathcal{W}}_{\mathrm{SW}}(\mu_t^{f_t}, \mu_v^{f_v}) + \zeta E(\phi) \tag{18}$$

where the excess cost function is

$$E(\phi) = \phi \cdot \bar{d}(\mathbf{z}, y_z) + 2\kappa(\phi \log \phi - \phi + 1) \tag{19}$$

To find the tightest upper bound, we minimize $E(\phi)$ over $\phi \in [0, 1]$ by taking the derivative:

$$\frac{dE}{d\phi} = \bar{d}(\mathbf{z}, y_z) + 2\kappa \log \phi \tag{20}$$

Setting this equal to zero yields the optimal parameter $\phi^* = e^{-\bar{d}(\mathbf{z}, y_z)/(2\kappa)}$.

Substituting the optimal parameter $\phi^*$ into the excess cost function and simplifying, we get

$$E(\phi^*) = e^{-\bar{d}/(2\kappa)} \cdot \bar{d} + 2\kappa \left( e^{-\bar{d}/(2\kappa)} \cdot \left( -\frac{\bar{d}}{2\kappa} \right) - e^{-\bar{d}/(2\kappa)} + 1 \right) \tag{21}$$

$$= 2\kappa \left( 1 - e^{-\bar{d}(\mathbf{z}, y_z)/(2\kappa)} \right) \tag{22}$$

Therefore, we conclude that

$$\tilde{\mathcal{W}}_{\mathrm{SW}}(\tilde{\mu}_t^{f_t}, \mu_v^{f_v}) \le (1-\zeta)\tilde{\mathcal{W}}_{\mathrm{SW}}(\mu_t^{f_t}, \mu_v^{f_v}) + 2\zeta\kappa \left( 1 - e^{-\bar{d}(\mathbf{z}, y_z)/(2\kappa)} \right) \tag{23}$$

This bound demonstrates the robustness of the $\mathcal{W}_{\mathrm{SW}}$ distance: the impact of the outlier is modulated by both its proportion $\zeta$ and its distance to the validation set, with the regularization parameter $\kappa$ controlling how aggressively distant outliers are penalized through the selective matching mechanism. $\qquad\square$

**Empirical Validation of Theorem 5.1.** To demonstrate that the distribution patterns of data value scores can distinguish isolated anomalies from systemic regime shifts, we designed a controlled synthetic experiment. This approach provides complete control over ground truth, enabling precise validation of the theoretical predictions from Theorem 5.1.

We generated a clean reference time series, $x(t) = \sin(2\pi t/100) + 0.5\sin(2\pi t/25) + \epsilon, \epsilon \sim \mathcal{N}(0, 0.01)$. We then constructed two test datasets. **Dataset A (Isolated Anomalies)** contains 50 sporadic spike anomalies (5% contamination) with magnitudes between 8 and 12 standard deviations at the time of injection, resulting in progressively larger absolute magnitudes as the contamination accumulates. This creates a realistic cascading effect often observed in deteriorating sensor systems where initial faults compound over time. **Dataset B (Regime Shift)** exhibits a systematic change at the midpoint, where the second half has an altered frequency and amplitude, simulating operational reconfiguration.

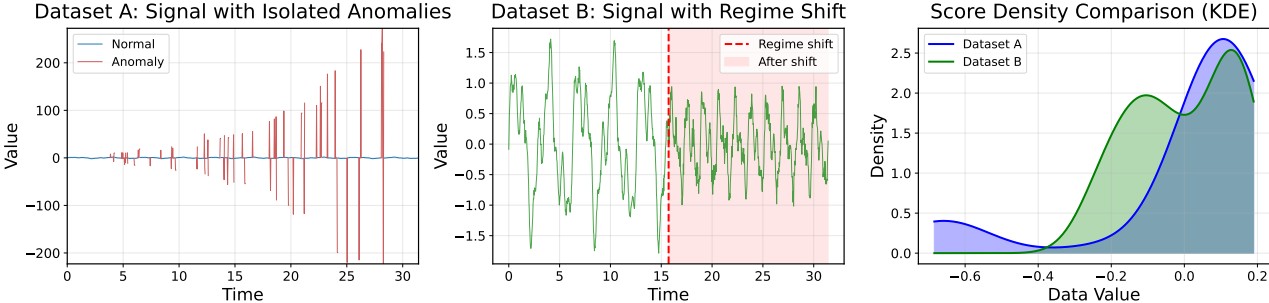

*Figure 7.* Experimental validation of Theorem 5.1: TIMELAVA distinguishes between isolated anomalies and regime shifts through characteristic score distributions. **Left**: Isolated anomalies manifest as sporadic spikes distributed across the time series (red markers indicate 50 anomalies, 5% of data). **Middle**: Regime shift exhibits systematic change after the transition point (red dashed line at $t = 500$), with the shaded region indicating the new regime. **Right**: Score density comparison reveals fundamentally different distributions: isolated anomalies produce a heavy-tailed distribution with extreme outliers (blue), while regime shifts yield a bimodal distribution (green) reflecting pre- and post-shift regimes. The JS divergence of $0.47$ confirms statistical distinguishability, demonstrating that the bounded response property of the $\mathcal{W}_{\text{SW}}$ distance enables nuanced data quality assessment.

As shown in Figure 7, the induced score distributions are fundamentally different. Isolated anomalies yield a heavy-tailed unimodal distribution (kurtosis = 2.95, skewness = $-2.22$; minimum = $-0.67$), whereas the regime shift produces a bimodal distribution with two peaks corresponding to pre/post-shift regimes and no extreme values (bimodality score = 0.38; range = $[-0.23, 0.19]$). The Jensen–Shannon divergence of $0.47$ further confirms statistical distinguishability. This pattern is consistent with the bound in Eq. (23), which encodes a trade-off between anomaly severity $\bar{d}$ and prevalence $\zeta$: large $\bar{d}$ but small $\zeta$ pushes a few segments close to the bound, producing heavy-tailed distribution with extreme outliers, while moderate $\bar{d}$ with large $\zeta$ shifts many segments coherently, yielding bimodal distributions reflecting dual regimes.

A key insight from our experiments is that, as established by Theorem 5.1, the bounded-response property naturally induces distinguishable distribution patterns for different anomaly types, enabling actionable data curation without hand-crafted design. Consequently, a single scoring rule enables both detection and differentiation of anomaly types without hand-crafted detectors. Consistent real-world behavior is observed in Fig. 14, where TIMELAVA-OT shows sharp, isolated negative peaks for true anomalies in SMAP channels G-1 and A-2 while remaining stable elsewhere.

### B.3. Proof of Theorem 5.3: Validation Performance Bound

*Proof.* We adapt the proof structure from (Just et al., 2023) for the wavelet distance $d_{\text{wav}}$ and the $\mathcal{W}_{\text{SW}}$ cost $\mathbf{D}$. Let $\Delta L$ denote the difference in expected loss between validation and training:

$$\Delta L = \mathbb{E}_{(\mathbf{x},y) \sim \mu_v^{f_v}}\Big[ L\big(y, f(\mathbf{x})\big)\Big] - \mathbb{E}_{(\mathbf{x},y) \sim \mu_t^{f_t}}\Big[ L\big(y, f(\mathbf{x})\big)\Big]. \tag{24}$$

Let $\mathcal{Z} = \mathcal{X} \times \mathcal{Y}$ be the joint space, and let $\pi^*$ be the optimal coupling that achieves the minimum in $\mathcal{W}_{HW}\big(\mu_t^{f_t}, \mu_v^{f_v}\big)$. Since $\pi^*$ is a coupling between $\mu_t^{f_t}$ and $\mu_v^{f_v}$, we can express:

$$\begin{aligned}
\Delta L &= \int_{\mathcal{Z}^2} \big[ L(y_v, f(\mathbf{x}_v)) - L(y_t, f(\mathbf{x}_t)) \big]\, d\pi^*((\mathbf{x}_t, y_t), (\mathbf{x}_v, y_v)) \\
&= \int_{\mathcal{Z}^2} \big[ L(y_v, f(\mathbf{x}_v)) - L(y_v, f(\mathbf{x}_t)) + L(y_v, f(\mathbf{x}_t)) - L(y_t, f(\mathbf{x}_t)) \big]\, d\pi^* \\
&\le \underbrace{\int_{\mathcal{Z}^2} \big| L(y_v, f(\mathbf{x}_v)) - L(y_v, f(\mathbf{x}_t)) \big|\, d\pi^*}_{U_1} + \underbrace{\int_{\mathcal{Z}^2} \big| L(y_v, f(\mathbf{x}_t)) - L(y_t, f(\mathbf{x}_t)) \big|\, d\pi^*}_{U_2}
\end{aligned} \tag{25}$$

where the last inequality follows from the triangle inequality.

**Bounding $U_1$:** Using the Lipschitz properties of $L$ and $f$:

$$U_1 \leq \int_{\mathcal{Z}^2} k|f(\mathbf{x}_v) - f(\mathbf{x}_t)| \, d\pi^*((\mathbf{x}_t, y_t), (\mathbf{x}_v, y_v)) \quad (L \text{ is } k\text{-Lipschitz})$$

$$\leq k\epsilon \int_{\mathcal{Z}^2} d_{\text{wav}}(\mathbf{x}_v, \mathbf{x}_t) \, d\pi^*((\mathbf{x}_t, y_t), (\mathbf{x}_v, y_v)) \quad (f \text{ is } \epsilon\text{-Lipschitz w.r.t. } d_{\text{wav}}). \tag{26}$$

**Bounding $U_2$:** Using the $k$-Lipschitz property of $L$ in its first argument:

$$U_2 \leq \int_{\mathcal{Z}^2} k|y_v - y_t| \, d\pi^*((\mathbf{x}_t, y_t), (\mathbf{x}_v, y_v)). \tag{27}$$

Let $\pi^*_{\mathcal{X} \times \mathcal{X}}$ denote the marginal distribution of $\pi^*$ on $\mathcal{X} \times \mathcal{X}$:

$$\pi^*_{\mathcal{X} \times \mathcal{X}}(\mathbf{x}_t, \mathbf{x}_v) = \int_{\mathcal{Y}^2} d\pi^*((\mathbf{x}_t, y_t), (\mathbf{x}_v, y_v)). \tag{28}$$

Define the violation set:

$$V = \{(\mathbf{x}_t, \mathbf{x}_v) \in \mathcal{X} \times \mathcal{X} : |f_v(\mathbf{x}_v) - f_t(\mathbf{x}_t)| > \epsilon_{tv} \cdot d_{\text{wav}}(\mathbf{x}_t, \mathbf{x}_v)\}. \tag{29}$$

By assumption (iv) and Definition 5.2:

$$\pi^*_{\mathcal{X} \times \mathcal{X}}(V) = \mathbb{P}_{(\mathbf{x}_t, \mathbf{x}_v) \sim \pi^*_{\mathcal{X} \times \mathcal{X}}}[|f_v(\mathbf{x}_v) - f_t(\mathbf{x}_t)| > \epsilon_{tv} \cdot d_{\text{wav}}(\mathbf{x}_t, \mathbf{x}_v)] \leq \delta_{\text{wav}}. \tag{30}$$

Define the extended violation set in $\mathcal{Z}^2$:

$$\tilde{V} = \{((\mathbf{x}_t, y_t), (\mathbf{x}_v, y_v)) \in \mathcal{Z}^2 : (\mathbf{x}_t, \mathbf{x}_v) \in V\}. \tag{31}$$

Since $y_t = f_t(\mathbf{x}_t)$ and $y_v = f_v(\mathbf{x}_v)$ for the support of $\pi^*$:

$$\pi^*(\tilde{V}) = \pi^*_{\mathcal{X} \times \mathcal{X}}(V) \leq \delta_{\text{wav}}. \tag{32}$$

We can decompose $U_2$:

$$U_2 \leq \int_{\tilde{V}} k|y_v - y_t| \, d\pi^* + \int_{\mathcal{Z}^2 \setminus \tilde{V}} k|y_v - y_t| \, d\pi^*$$

$$\leq \int_{\tilde{V}} k \cdot 2M \, d\pi^* + \int_{\mathcal{Z}^2 \setminus \tilde{V}} k \cdot \epsilon_{tv} d_{\text{wav}}(\mathbf{x}_t, \mathbf{x}_v) \, d\pi^*$$

$$(\text{since } |y_v|, |y_t| \leq M \text{ and on } \mathcal{Z}^2 \setminus \tilde{V}: |y_v - y_t| \leq \epsilon_{tv} d_{\text{wav}}(\mathbf{x}_t, \mathbf{x}_v))$$

$$\leq 2kM \cdot \pi^*(\tilde{V}) + k\epsilon_{tv} \int_{\mathcal{Z}^2} d_{\text{wav}}(\mathbf{x}_t, \mathbf{x}_v) \, d\pi^*$$

$$\leq 2kM\delta_{\text{wav}} + k\epsilon_{tv} \int_{\mathcal{Z}^2} d_{\text{wav}}(\mathbf{x}_t, \mathbf{x}_v) \, d\pi^*. \tag{33}$$

**Combining the bounds:** From Eqs. (26) and (33):

$$\Delta L \leq U_1 + U_2$$

$$\leq k\epsilon \int_{\mathcal{Z}^2} d_{\text{wav}}(\mathbf{x}_t, \mathbf{x}_v) \, d\pi^* + 2kM\delta_{\text{wav}} + k\epsilon_{tv} \int_{\mathcal{Z}^2} d_{\text{wav}}(\mathbf{x}_t, \mathbf{x}_v) \, d\pi^*$$

$$= k(\epsilon + \epsilon_{tv}) \int_{\mathcal{Z}^2} d_{\text{wav}}(\mathbf{x}_t, \mathbf{x}_v) \, d\pi^* + 2kM\delta_{\text{wav}}. \tag{34}$$

We now need to relate the integral $\int_{\mathcal{Z}^2} d_{\text{wav}}(\mathbf{x}_t, \mathbf{x}_v) d\pi^*$ to $\mathcal{W}_{\text{SW}}(\mu_t^{f_t}, \mu_v^{f_v})$. Let $\pi^*$ be the optimal coupling for $\mathcal{W}_{\text{SW}}$, i.e., the optimal transport plan that solves the above optimization problem. Consider the expected cost under this optimal plan:

$$\mathbb{E}_{\pi^*}[\mathbf{D}((\mathbf{x}_t, y_t), (\mathbf{x}_v, y_v))] = \int_{\mathcal{Z}^2} [d_{\text{wav}}(\mathbf{x}_t, \mathbf{x}_v) + c \cdot \mathcal{W}_{d_{\text{wav}}}(\mu_t(\cdot|y_t), \mu_v(\cdot|y_v))] d\pi^*$$

$$= \int_{\mathcal{Z}^2} d_{\text{wav}}(\mathbf{x}_t, \mathbf{x}_v) d\pi^* + c \cdot \int_{\mathcal{Z}^2} \mathcal{W}_{d_{\text{wav}}}(\mu_t(\cdot|y_t), \mu_v(\cdot|y_v)) d\pi^* \tag{35}$$

Since the second term is non-negative (as the Wasserstein distance is non-negative), we have:

$$\int_{\mathcal{Z}^2} d_{\mathrm{wav}}(\mathbf{x}_t, \mathbf{x}_v) d\pi^* \leq \mathbb{E}_{\pi^*}[\mathbf{D}((\mathbf{x}_t, y_t), (\mathbf{x}_v, y_v))] \tag{36}$$

Now, since $\pi^*$ is the optimal transport plan for $\mathcal{W}_{\mathrm{SW}}$, we have:

$$\mathbb{E}_{\pi^*}[\mathbf{D}((\mathbf{x}_t, y_t), (\mathbf{x}_v, y_v))] = \langle \mathbf{D}, \pi^* \rangle \tag{37}$$

where $\langle \mathbf{D}, \pi^* \rangle$ denotes the Frobenius inner product between the cost matrix and transport plan.

Considering the contribution of the KL divergence terms at the optimal solution $\pi^*$, we have:

$$\mathcal{W}_{\mathrm{SW}}(\mu_t^{f_t}, \mu_v^{f_v}) = \langle \mathbf{D}, \pi^* \rangle + \kappa D_{\mathrm{KL}}(\pi^* \mathbf{1}_m \| \boldsymbol{\Delta}_n) + \kappa D_{\mathrm{KL}}((\pi^*)^\top \mathbf{1}_n \| \boldsymbol{\Delta}_m) \tag{38}$$

This implies:

$$\langle \mathbf{D}, \pi^* \rangle \leq \mathcal{W}_{\mathrm{SW}}(\mu_t^{f_t}, \mu_v^{f_v}) \tag{39}$$

since the KL divergence is non-negative, so $\kappa D_{\mathrm{KL}}(\pi^* \mathbf{1}_m \| \boldsymbol{\Delta}_n) + \kappa D_{\mathrm{KL}}((\pi^*)^\top \mathbf{1}_n \| \boldsymbol{\Delta}_m) \geq 0$.

Combining these relationships:

$$\int_{\mathcal{Z}^2} d_{\mathrm{wav}}(\mathbf{x}_t, \mathbf{x}_v) d\pi^* \leq \mathbb{E}_{\pi^*}[\mathbf{D}((\mathbf{x}_t, y_t), (\mathbf{x}_v, y_v))] = \langle \mathbf{D}, \pi^* \rangle \leq \mathcal{W}_{\mathrm{SW}}(\mu_t^{f_t}, \mu_v^{f_v}) \tag{40}$$

Therefore:

$$\int_{\mathcal{Z}^2} d_{\mathrm{wav}}(\mathbf{x}_t, \mathbf{x}_v) d\pi^* \leq \mathcal{W}_{\mathrm{SW}}(\mu_t^{f_t}, \mu_v^{f_v}) \tag{41}$$

Furthermore, in our derivation, if we assume that $\epsilon_{tv} \leq c\epsilon$ (i.e., the cross-Lipschitz constant is related to the hierarchical cost weight $c$ and the model Lipschitz constant $\epsilon$), then $\epsilon + \epsilon_{tv} \leq \epsilon(1 + c)$. This allows us to further simplify the upper bound:

$$\Delta L \leq k(\epsilon + \epsilon_{tv}) \int_{\mathcal{Z}^2} d_{\mathrm{wav}}(\mathbf{x}_t, \mathbf{x}_v) d\pi^* + 2kM\delta_{\mathrm{wav}} \tag{42}$$

$$\leq k\epsilon(1 + c) \int_{\mathcal{Z}^2} d_{\mathrm{wav}}(\mathbf{x}_t, \mathbf{x}_v) d\pi^* + 2kM\delta_{\mathrm{wav}} \tag{43}$$

$$\leq k\epsilon(1 + c)\mathcal{W}_{\mathrm{SW}}(\mu_t^{f_t}, \mu_v^{f_v}) + 2kM\delta_{\mathrm{wav}} \tag{44}$$

For simplicity of notation, we absorb the constant $(1 + c)$ into $\epsilon$, and write $k\epsilon(1 + c)$ as $k\epsilon$. This leads to:

$$\Delta L \leq k\epsilon \cdot \mathcal{W}_{\mathrm{SW}}(\mu_t^{f_t}, \mu_v^{f_v}) + 2kM\delta_{\mathrm{wav}} \tag{45}$$

This completes the derivation, showing that the gap between validation and training performance is bounded by the $\mathcal{W}_{\mathrm{SW}}$ distance, which is our main result. $\qquad\square$

## B.4. Proof of Theorem 4.5: Data Value Sensitivity

The proof proceeds in three main steps: establishing the dual formulation, proving uniqueness of the optimal solution, and deriving the properties of data values.

*Proof.* We begin with the primal unbalanced optimal transport problem. Using the KL divergence formulation $D_{\mathrm{KL}}(p\|q) = \sum_i p_i \log(p_i/q_i) - p_i + q_i$, the primal problem becomes

$$\min_{\mathbf{T} \geq 0} \left\{ \langle \mathbf{D}, \mathbf{T} \rangle + \kappa \sum_{i=1}^n h\left(\frac{(\mathbf{T}\mathbf{1}_m)_i}{1/n}\right) + \kappa \sum_{j=1}^m h\left(\frac{(\mathbf{T}^\top \mathbf{1}_n)_j}{1/m}\right) \right\} \tag{46}$$

where $h(x) = x \log x - x + 1$ is the generalized KL divergence kernel. To derive the dual, we apply the Fenchel-Rockafellar duality theorem with the Lagrangian

$$\mathcal{L}(\mathbf{T}, \lambda) = \langle \mathbf{D}, \mathbf{T} \rangle + \kappa \sum_{i=1}^{n} h\left(\frac{(\mathbf{T}\mathbf{1}_m)_i}{1/n}\right) + \kappa \sum_{j=1}^{m} h\left(\frac{(\mathbf{T}^\top \mathbf{1}_n)_j}{1/m}\right) - \langle \lambda, \mathbf{T} \rangle. \tag{47}$$

Since the Fenchel conjugate of $h$ is $h^*(y) = e^y - 1$, we can apply the minimax theorem. Taking the gradient with respect to $\mathbf{T}$ and setting it to zero yields

$$\mathbf{D}_{ij} + \kappa \log\left(n(\mathbf{T}\mathbf{1}_m)_i\right) + \kappa \log\left(m(\mathbf{T}^\top \mathbf{1}_n)_j\right) = \lambda_{ij}. \tag{48}$$

Setting $\mathbf{f}_i = -\kappa \log\left(n(\mathbf{T}\mathbf{1}_m)_i\right)$ and $\mathbf{g}_j = -\kappa \log\left(m(\mathbf{T}^\top \mathbf{1}_n)_j\right)$, we obtain $\lambda_{ij} = \mathbf{D}_{ij} - \mathbf{f}_i - \mathbf{g}_j$. For $\mathbf{T} \geq 0$, we require $\lambda_{ij} \geq 0$, yielding the constraint $\mathbf{f}_i + \mathbf{g}_j \leq \mathbf{D}_{ij}$.

Substituting back into the dual objective and using $\psi_\kappa(u) = \kappa(1 - e^{-u/\kappa})$ as the Fenchel conjugate of the scaled entropy, we obtain the dual formulation:

$$\mathcal{W}_{\text{SW}}(\mu_t, \mu_v) = \max_{\mathbf{f}, \mathbf{g}} \sum_{i=1}^{n} \frac{\kappa}{n} \psi_\kappa(\mathbf{f}_i) + \sum_{j=1}^{m} \frac{\kappa}{m} \psi_\kappa(\mathbf{g}_j) \tag{49}$$

subject to $\mathbf{f}_i + \mathbf{g}_j \leq \mathbf{D}_{ij}$ for all $i, j$.

To establish uniqueness of the optimal solution, we analyze the dual objective function

$$L(\mathbf{f}, \mathbf{g}) = \sum_{i=1}^{n} \frac{\kappa}{n} \psi_\kappa(\mathbf{f}_i) + \sum_{j=1}^{m} \frac{\kappa}{m} \psi_\kappa(\mathbf{g}_j). \tag{50}$$

The second derivative of $\psi_\kappa$ is $\frac{d^2 \psi_\kappa(u)}{du^2} = -\frac{1}{\kappa} e^{-u/\kappa} < 0$, showing that $\psi_\kappa$ is strictly concave. Therefore, the objective function $L(\mathbf{f}, \mathbf{g})$ is strictly concave in $(\mathbf{f}, \mathbf{g})$, while the constraint set defined by linear inequalities is convex. By the fundamental theorem of convex optimization, the optimal dual solution $(\mathbf{f}^*, \mathbf{g}^*)$ is unique and satisfies the KKT conditions: $T_{ij}^* > 0 \Rightarrow \mathbf{f}_i^* + \mathbf{g}_j^* = \mathbf{D}_{ij}$, along with the primal-dual relationships $\sum_{j=1}^{m} T_{ij}^* = \frac{1}{n} e^{-\mathbf{f}_i^*/\kappa}$ and $\sum_{i=1}^{n} T_{ij}^* = \frac{1}{m} e^{-\mathbf{g}_j^*/\kappa}$.

Given the unique optimal dual solution, we define the relative data value as

$$v(\mathbf{x}_i) = -\left[\psi_\kappa(\mathbf{f}_i^*) - \frac{1}{n-1} \sum_{j \neq i} \psi_\kappa(\mathbf{f}_j^*)\right] \tag{51}$$

The uniqueness and deterministic ranking properties follow directly from the uniqueness of $(\mathbf{f}^*, \mathbf{g}^*)$ and the strict monotonicity of $\psi_\kappa$. For interpretability, $v(\mathbf{x}_i) > 0$ indicates that $\psi_\kappa(\mathbf{f}_i^*)$ is below the average value of other samples, suggesting that this sample helps align the distributions (lower transport cost).

To establish the sensitivity interpretation, we consider a mass perturbation around sample $i$:

$$\mu^\delta = \left(\frac{1}{n} + \delta\right) \delta_{\mathbf{x}_i} + \sum_{j \neq i} \left(\frac{1}{n} - \frac{\delta}{n-1}\right) \delta_{\mathbf{x}_j} \tag{52}$$

which preserves total mass. The directional derivative of $\mathcal{W}_{\text{SW}}$ with respect to this perturbation at $\delta = 0$ is

$$\frac{d}{d\delta} \mathcal{W}_{\text{SW}}(\mu^\delta, \mu_v)\bigg|_{\delta=0} = \frac{\partial \mathcal{W}_{\text{SW}}}{\partial \mu_i} - \frac{1}{n-1} \sum_{j \neq i} \frac{\partial \mathcal{W}_{\text{SW}}}{\partial \mu_j}. \tag{53}$$

By the envelope theorem applied to the dual formulation, $\frac{\partial \mathcal{W}_{\text{SW}}}{\partial \mu_i} = \frac{\kappa}{n} \psi_\kappa(\mathbf{f}_i^*)$, which yields

$$\frac{d}{d\delta} \mathcal{W}_{\text{SW}}(\mu^\delta, \mu_v)\bigg|_{\delta=0} = \frac{\kappa}{n}\left[\psi_\kappa(\mathbf{f}_i^*) - \frac{1}{n-1} \sum_{j \neq i} \psi_\kappa(\mathbf{f}_j^*)\right] = -\frac{\kappa}{n} v(\mathbf{x}_i). \tag{54}$$

This shows that $v(\mathbf{x}_i)$ measures the sensitivity of the $\mathcal{W}_{\text{SW}}$ distance to local mass perturbations at sample $i$, providing an economic interpretation of data value in terms of marginal contribution to distributional alignment. $\square$

## B.5. Proof of Theorem B.2: Convergence of Entropy-Regularized Approximation

**Theorem B.2** (Convergence of Entropy-Regularized Approximation). *Let $v(\mathbf{x}_i)$ and $v_\varepsilon(\mathbf{x}_i)$ be the data values derived from the optimal dual potentials $(\mathbf{f}^*, \mathbf{g}^*)$ and $(\mathbf{f}_\varepsilon^*, \mathbf{g}_\varepsilon^*)$ of the unregularized and entropy-regularized $\mathcal{W}_{SW}$ problems, respectively. Then:*

1. ***Pointwise Convergence:*** *As $\varepsilon \to 0$, we have $v_\varepsilon(\mathbf{x}_i) \to v(\mathbf{x}_i)$ for all $i \in [n]$.*

2. ***Ranking Preservation:*** *For any pair of time series segments such that $v(\mathbf{x}_i) > v(\mathbf{x}_j)$, there exists an $\varepsilon_0 > 0$ such that for all $0 < \varepsilon < \varepsilon_0$, the ranking is preserved: $v_\varepsilon(\mathbf{x}_i) > v_\varepsilon(\mathbf{x}_j)$.*

*Proof.* The proof builds on the convergence theory for unbalanced optimal transport (Séjourné et al., 2019; Pham et al., 2020). The data values are defined as functions of the optimal dual potentials:

$$v(\mathbf{x}_i) = \psi_\kappa(\mathbf{f}_i^*) - \frac{1}{n-1} \sum_{j \neq i} \psi_\kappa(\mathbf{f}_j^*) \tag{55}$$

$$v_\varepsilon(\mathbf{x}_i) = \psi_\kappa(\mathbf{f}_{\varepsilon,i}^*) - \frac{1}{n-1} \sum_{j \neq i} \psi_\kappa(\mathbf{f}_{\varepsilon,j}^*) \tag{56}$$

where $\psi_\kappa(u) = \kappa(1 - e^{-u/\kappa})$ is the transformation function arising from the KL divergence penalty.

Following Séjourné et al. (2019) (Proposition 21), the optimal dual potentials are uniformly bounded: there exists $M > 0$ independent of $\varepsilon$ such that $\|\mathbf{f}^*\|_\infty, \|\mathbf{f}_\varepsilon^*\|_\infty \leq M$ and $\|\mathbf{g}^*\|_\infty, \|\mathbf{g}_\varepsilon^*\|_\infty \leq M$. This boundedness is established through the coercivity of the dual functional and holds for unbalanced optimal transport with KL divergence penalties on compact domains.

**Convergence of Dual Potentials.** From Séjourné et al. (2019) (Proposition 10), for the unbalanced optimal transport setting with KL divergence penalties, under the assumptions of compact domain $\mathcal{X}$ and Lipschitz continuous cost $D^{(W)}$, the optimal dual potentials converge uniformly as the regularization parameter vanishes:

$$\lim_{\varepsilon \to 0} \|\mathbf{f}_\varepsilon^* - \mathbf{f}^*\|_\infty = 0 \quad \text{and} \quad \lim_{\varepsilon \to 0} \|\mathbf{g}_\varepsilon^* - \mathbf{g}^*\|_\infty = 0 \tag{57}$$

This convergence is established through weak continuous dependence of dual potentials on the regularization parameter. Furthermore, Séjourné et al. (2019) (Theorem 1) guarantees that for any fixed $\varepsilon > 0$, the optimal potentials $(\mathbf{f}_\varepsilon^*, \mathbf{g}_\varepsilon^*)$ can be computed via the Sinkhorn algorithm with linear convergence.

**Convergence of Data Values.** The function $\psi_\kappa(u) = \kappa(1 - e^{-u/\kappa})$ has derivative $\psi_\kappa'(u) = e^{-u/\kappa}$. Since the dual potentials are uniformly bounded in $[-M, M]$, $\psi_\kappa$ is Lipschitz continuous on this compact set with Lipschitz constant:

$$L = \sup_{u \in [-M,M]} |\psi_\kappa'(u)| = \sup_{u \in [-M,M]} e^{-u/\kappa} = e^{M/\kappa} < \infty \tag{58}$$

Using this Lipschitz property and the triangle inequality:

$$|v_\varepsilon(\mathbf{x}_i) - v(\mathbf{x}_i)| = \left| \left( \psi_\kappa(\mathbf{f}_{\varepsilon,i}^*) - \psi_\kappa(\mathbf{f}_i^*) \right) - \frac{1}{n-1} \sum_{j \neq i} \left( \psi_\kappa(\mathbf{f}_{\varepsilon,j}^*) - \psi_\kappa(\mathbf{f}_j^*) \right) \right| \tag{59}$$

$$\leq |\psi_\kappa(\mathbf{f}_{\varepsilon,i}^*) - \psi_\kappa(\mathbf{f}_i^*)| + \frac{1}{n-1} \sum_{j \neq i} |\psi_\kappa(\mathbf{f}_{\varepsilon,j}^*) - \psi_\kappa(\mathbf{f}_j^*)| \tag{60}$$

$$\leq L|\mathbf{f}_{\varepsilon,i}^* - \mathbf{f}_i^*| + \frac{L}{n-1} \sum_{j \neq i} |\mathbf{f}_{\varepsilon,j}^* - \mathbf{f}_j^*| \tag{61}$$

$$\leq L\|\mathbf{f}_\varepsilon^* - \mathbf{f}^*\|_\infty + L\|\mathbf{f}_\varepsilon^* - \mathbf{f}^*\|_\infty \tag{62}$$

$$= 2L\|\mathbf{f}_\varepsilon^* - \mathbf{f}^*\|_\infty \tag{63}$$

From above, we know that $\|\mathbf{f}_\varepsilon^* - \mathbf{f}^*\|_\infty \to 0$ as $\varepsilon \to 0$. Since $L$ is a finite constant (independent of $\varepsilon$), we obtain:

$$\lim_{\varepsilon \to 0} |v_\varepsilon(\mathbf{x}_i) - v(\mathbf{x}_i)| = \lim_{\varepsilon \to 0} 2L\|\mathbf{f}_\varepsilon^* - \mathbf{f}^*\|_\infty = 0 \quad \text{for all } i \in [n] \tag{64}$$

which establishes pointwise convergence.

**Ranking Preservation.** Suppose $v(\mathbf{x}_i) > v(\mathbf{x}_j)$ and let $\delta = v(\mathbf{x}_i) - v(\mathbf{x}_j) > 0$ be the gap between these data values. From the pointwise convergence established in Step 3, for the positive value $\delta/3$, there exists an $\varepsilon_0 > 0$ such that for all $0 < \varepsilon < \varepsilon_0$:

$$|v_\varepsilon(\mathbf{x}_k) - v(\mathbf{x}_k)| < \frac{\delta}{3} \quad \text{for all } k \in \{i, j\} \tag{65}$$

Using this bound:

$$v_\varepsilon(\mathbf{x}_i) - v_\varepsilon(\mathbf{x}_j) = (v(\mathbf{x}_i) - v(\mathbf{x}_j)) + (v_\varepsilon(\mathbf{x}_i) - v(\mathbf{x}_i)) - (v_\varepsilon(\mathbf{x}_j) - v(\mathbf{x}_j)) \tag{66}$$

$$= \delta + (v_\varepsilon(\mathbf{x}_i) - v(\mathbf{x}_i)) - (v_\varepsilon(\mathbf{x}_j) - v(\mathbf{x}_j)) \tag{67}$$

$$\geq \delta - |v_\varepsilon(\mathbf{x}_i) - v(\mathbf{x}_i)| - |v_\varepsilon(\mathbf{x}_j) - v(\mathbf{x}_j)| \tag{68}$$

$$> \delta - \frac{\delta}{3} - \frac{\delta}{3} \tag{69}$$

$$= \frac{\delta}{3} > 0 \tag{70}$$

Therefore, for all $0 < \varepsilon < \varepsilon_0$, we have $v_\varepsilon(\mathbf{x}_i) > v_\varepsilon(\mathbf{x}_j)$, which proves that the ranking is preserved. $\qquad\square$

**Numerical Verification of Theorem B.2.** We verify the convergence properties using synthetic time series data. We generate 40 training time series and 30 validation time series, each segmented into overlapping windows. This yields training segments labeled as high quality (clean sinusoids, $\sigma = 0.05$), medium quality (moderate noise with harmonics, $\sigma = 0.2$), or low quality (pure noise or severe outliers). We compute segment-wise data values using $\varepsilon \in [10^{-4}, 1]$ with 20 logarithmically-spaced values and $\kappa = 1.0$.

Fig.8 presents three verification aspects. The left plot shows segment value trajectories for 10 representative segments, demonstrating pointwise convergence as $\varepsilon \to 0$. The middle plot displays $L_1$ and $L_\infty$ errors between $v_\varepsilon$ and the reference solution ($\varepsilon_{\text{ref}} = 10^{-4}$) on a log-log scale. The empirical convergence rate of $O(\varepsilon^{0.88})$ closely matches the theoretical $O(\varepsilon)$ prediction, with the $L_\infty$ error naturally exceeding $L_1$ as it captures worst-case segment behavior. The right plot shows Spearman correlation coefficients consistently above 0.95, confirming that segment rankings remain stable across all $\varepsilon$ values. These results validate both claims of Theorem B.2: (i) segment-wise convergence $v_\varepsilon(\mathbf{x}_i) \to v(\mathbf{x}_i)$ as $\varepsilon \to 0$, and (ii) ranking preservation across the full range of regularization parameters.

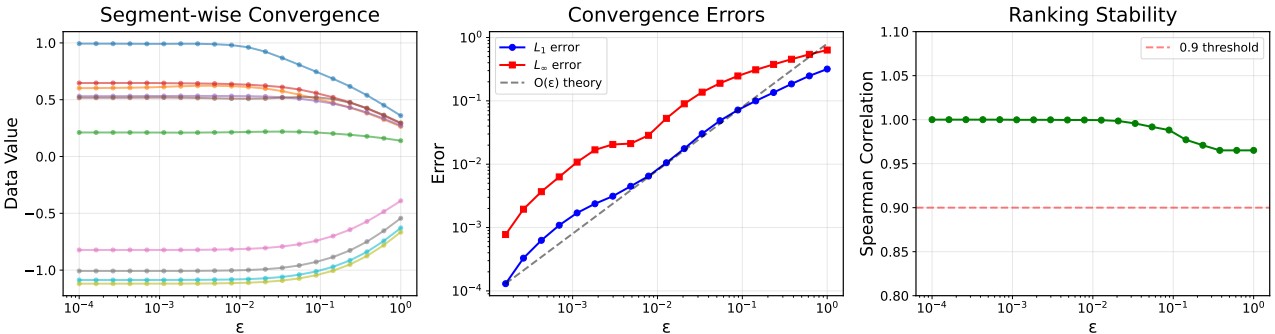

*Figure 8.* Numerical verification of entropy-regularized UOT convergence for time series segments. Left: Value trajectories showing segment-wise convergence. Middle: $L_1$ and $L_\infty$ convergence errors on log-log scale with theoretical $O(\varepsilon)$ reference (dashed). Right: Spearman correlation demonstrating ranking stability.

# C. Experiments

## C.1. The TimeLAVA Algorithm

In this appendix, we present the segment-wise and point-wise valuation algorithms for time series, which together complete the TimeLAVA framework. We also provide an analysis of the computational complexity of the algorithm and clarify the role of the reference time series.

C.1.1. SEGMENT-WISE VALUATION ALGORITHM

Building on the methodology developed, the detailed segment-wise valuation procedure of TIMELAVA is summarized in Algorithm 1.

---

**Algorithm 1** TIMELAVA Algorithm

---

**Require:** Evaluated segments $\mathcal{D}_{\text{eval}} = \{(\mathbf{x}_i, y_i)\}_{i=1}^n$, reference segments $\mathcal{D}_{\text{ref}} = \{(\mathbf{x}'_j, y'_j)\}_{j=1}^m$, wavelet $\psi$, parameters $(\kappa, c, \epsilon)$

**Ensure:** Segment values $\{v_\epsilon(\mathbf{x}_i)\}_{i=1}^n$

    // Compute Wavelet Representations

1: Compute DWT coefficients $\Psi(\mathbf{x}_i)$ for all $\mathbf{x}_i \in \mathcal{D}_{\text{eval}}$ and $\Psi(\mathbf{x}'_j)$ for all $\mathbf{x}'_j \in \mathcal{D}_{\text{ref}}$.

    // Compute Pairwise Cost Matrix $\mathbf{D}^{(W)}$

2: **for** $i = 1$ to $n$ **do**

3:     **for** $j = 1$ to $m$ **do**

4:         Compute ground wavelet distance $d_{\text{wav}}(\mathbf{x}_i, \mathbf{x}'_j) = \|\Psi(\mathbf{x}_i) - \Psi(\mathbf{x}'_j)\|_1$.

5:         **if** $c > 0$ **then**

6:             Compute $\mathcal{W}_{d_{\text{wav}}}\big(\mu_{\text{eval}}(\cdot|y_i), \mu_{\text{ref}}(\cdot|y'_j)\big)$ using $d_{\text{wav}}$ as the ground metric.

7:             Set $\mathbf{D}_{i,j}^{(W)} = d_{\text{wav}}(\mathbf{x}_i, \mathbf{x}'_j) + c \cdot \mathcal{W}_{d_{\text{wav}}}(\dots)$.

8:         **else**

9:             Set $\mathbf{D}_{i,j}^{(W)} = d_{\text{wav}}(\mathbf{x}_i, \mathbf{x}'_j)$.

10:         **end if**

11:     **end for**

12: **end for**

    // Solve Regularized UOT Problem

13: Solve the entropy-regularized UOT problem using Sinkhorn algorithm:

$$\min_{\mathbf{T} \geq 0} \langle \mathbf{D}^{(W)}, \mathbf{T} \rangle + \kappa D_{\text{KL}}(\mathbf{T}\mathbf{1}_m \| \boldsymbol{\Delta}_n) + \kappa D_{\text{KL}}(\mathbf{T}^\top \mathbf{1}_n \| \boldsymbol{\Delta}_m) - \epsilon H(\mathbf{T})$$

    to obtain the optimal dual variables $(\mathbf{f}_\epsilon^*, \mathbf{g}_\epsilon^*)$.

    // Compute Data Values

14: **for** $i = 1$ to $n$ **do**

15:     Compute $\phi_i = \psi_\kappa(\mathbf{f}_{\epsilon,i}^*) = \kappa(1 - e^{-\mathbf{f}_{\epsilon,i}^*/\kappa})$.

16: **end for**

17: Compute sum $S = \sum_{j=1}^n \phi_j$.

18: **for** $i = 1$ to $n$ **do**

19:     Compute $v_\epsilon(\mathbf{x}_i) = -\big(\phi_i - \frac{S - \phi_i}{n-1}\big)$.

20: **end for**

21: **return** $\{v_\epsilon(\mathbf{x}_i)\}_{i=1}^n$.

---

C.1.2. POINT-WISE VALUATION ALGORITHM

Algorithm 2 outlines the mapping process from segment contributions to point-wise contributions.

C.1.3. COMPUTATIONAL COMPLEXITY

The computational complexity of TIMELAVA is dominated by three components. First, the wavelet transform for $N$ training and $M$ validation segments of length $L$ requires $O((N + M) \cdot L)$ using Mallat's pyramidal algorithm. Second, computing the pairwise cost matrix requires $O(N \cdot M \cdot L)$ operations. Third, solving the entropy-regularized UOT problem requires $O(N \cdot M \cdot \log(1/\epsilon))$ operations. Together, the overall complexity is $O((N + M) \cdot L + N \cdot M \cdot L)$, which simplifies to $O(N \cdot M \cdot L)$ when $N \cdot M >> N + M$, as is typical in practice. When using label information ($c > 0$), an additional $O(N \cdot M \cdot K^2)$ term arises from computing conditional Wasserstein distances, where $K$ is the average number of segments per label class. All experiments were conducted on a workstation equipped with an Intel Xeon Gold 6448H CPU (2.0 GHz, 32 cores), and 512 GB RAM. While our implementation is CPU-based, it can achieve even faster runtimes with GPU acceleration. Unless otherwise specified, we set $\kappa = 2.0$ for the UOT regularization parameter, $\varepsilon = 0.01$ for the entropy

---

**Algorithm 2** Point-wise Contribution Calculation

---

**Require:** Segment contribution values $\{v_\epsilon(\mathbf{z}^{[n]})\}$, original time series length $N$, segment length $T$, sliding stride $s$
**Ensure:** Point-wise contribution values $\{v(t)\}_{t=1}^N$
 1: Initialize contribution counters $\text{count}[t] = 0$ and contribution sums $\text{sum}[t] = 0$ for each time point $t = 1, 2, \ldots, N$
 2: **for** each segment $\mathbf{z}^{[n]}$ and its contribution value $v_\epsilon(\mathbf{z}^{[n]})$ **do**
 3:     Determine the time range $[\text{start}, \text{end}]$ covered by this segment
 4:     **for** $t \in [\text{start}, \text{end}]$ **do**
 5:         $\text{sum}[t] \mathrel{+}= v_\epsilon(\mathbf{z}^{[n]})$
 6:         $\text{count}[t] \mathrel{+}= 1$
 7:     **end for**
 8: **end for**
 9: **for** each time point $t = 1, 2, \ldots, N$ **do**
10:     **if** $\text{count}[t] > 0$ **then**
11:         $v(t) = \text{sum}[t]/\text{count}[t]$
12:     **else**
13:         $v(t) = 0$ {Point not covered by any segment}
14:     **end if**
15: **end for**
16: **return** $\{v(t)\}_{t=1}^N$

---

regularization, and use a Daubechies-4 (db4) wavelet with decomposition level 2 across all experiments.

### C.1.4. ON THE ROLE OF REFERENCE TIME SERIES

In learning-agnostic valuation frameworks, the *reference distribution* is the distribution against which the contributions of training data are evaluated. In i.i.d. settings such as LAVA, the reference set is the validation set (Just et al., 2023). For time series applications, however, the choice of reference data depends on the task: validation sequences in forecasting, normal (non-anomalous) segments in anomaly detection, or clean segment–label pairs in label-noise detection. In our experiments, we adopt these task-specific settings when defining the reference time series. In all cases, the reference distribution provides a meaningful anchor, allowing valuation to quantify how much each training segment contributes to aligning the training distribution with the target task distribution.

### C.2. Anomaly Detection

Details of the datasets used in our anomaly detection experiments are summarized in Table 3. For each dataset, we report the dimensionality, number of time series, average length, and average anomaly ratio. We categorize the datasets into two groups:

1. **Pre-partitioned datasets** provide predefined splits consisting of a clean validation set, a contaminated test set with anomalies, and the corresponding labels (e.g., SMD (Su et al., 2019), SMAP and MSL (Hundman et al., 2018), PSM (Abdulaal et al., 2021), SWaT (Mathur & Tippenhauer, 2016)). We directly adopt these official splits for evaluating TIMELAVA.

2. **Single-set datasets** consist of a single continuous time series with anomaly labels only (e.g., WADI (Ahmed et al., 2017), NAB (Ahmad et al., 2017), KDD-CUP99 (Stolfo et al., 2000)). We manually partition each dataset into a clean validation set and a contaminated test set.

### C.2.1. ADDITIONAL EXPERIMENTAL RESULTS

Table 4 and Fig. 10 demonstrate TIMELAVA's consistently strong performance across all datasets, achieving the highest AUC on four of six benchmarks and leading F1 scores on NAB-Traffic and NAB-Tweets. This performance spans both univariate and multivariate anomaly detection tasks.

In contrast, TimeInf, despite being designed for time series valuation, shows limited effectiveness, particularly for local

*Table 3.* Details of the anomaly detection datasets. Note that all discrete-valued dimensions are excluded from each dataset. The term "Average Length" refers to the average length across all time series in the dataset.

| Dataset | Dimensions | Num. of Time Series | Average Length | Average Anomaly Ratio |
|---|---|---|---|---|
| UCR | 1 | 250 | 56,205 | 0.8% |
| SMAP | 25 | 55 | 8,068 | 12.8% |
| MSL | 55 | 27 | 2,730 | 10.5% |
| NAB-Traffic | 1 | 7 | 2,238 | 10.0% |
| SMD | 38 | 28 | 25,300 | 4.2% |
| NAB-AdExchange | 1 | 6 | 1,602 | 10.0% |
| NAB-Tweets | 1 | 10 | 15,863 | 9.9% |
| NAB-Taxi | 1 | 1 | 10,320 | 5.2% |
| PSM | 25 | 1 | 87,841 | 27.8% |
| SWaT | 51 | 1 | 449,919 | 12.1% |
| WADI | 123 | 1 | 172,751 | 5.7% |
| KDD-CUP99 | 34 | 1 | 494,021 | 19.7% |

contextual and point anomalies, as shown in Fig. 10. This weakness stems from TimeInf's reliance on influence functions, which assume approximate stationarity and struggle to capture highly localized temporal events. While influence functions excel at measuring global perturbation effects, they lack the time-frequency localization needed to identify brief, transient anomalies that manifest at specific temporal scales, a capability that is instead provided by TIMELAVA's wavelet-based approach. Moreover, TIMELAVA's performance benefits from larger datasets. Larger datasets provide more segment pairs for UOT matching, reducing valuation score variance and improving stability. This trend is evident across our benchmarks: as dataset size increases from SMAP (8k points, AUC=0.74) to PSM (87k, AUC=0.77) to SWaT (449k, AUC=0.86) to KDD-Cup99 (494k, AUC=0.98), performance consistently improves.

Additionally, our experimental setup intentionally restricts reference time series to approximately 2,000 consecutive time points for large datasets (e.g. PSM), reflecting realistic data-scarce scenarios that are common in practical deployments. This constraint particularly impacts deep learning methods: DCdetector and AnomalyTransformer require extensive parameterization, which depends on large-scale training data to learn robust temporal representations. When reference data is scarce, TIMELAVA maintains strong performance while deep learning baselines deteriorate significantly, demonstrating its superior robustness and practical applicability, a common challenge in real-world anomaly detection systems.

We have also included the comparison of anomaly score distributions between normal and anomalous segments to *visually* assess the discriminative capability of different methods (TIMELAVA, TimeInf, IsolationForest, and AnomalyTransformer) on the PSM dataset. As shown in Fig. 9, TIMELAVA demonstrates the clearest separation with minimal overlap, indicating superior discriminative power compared to other methods.

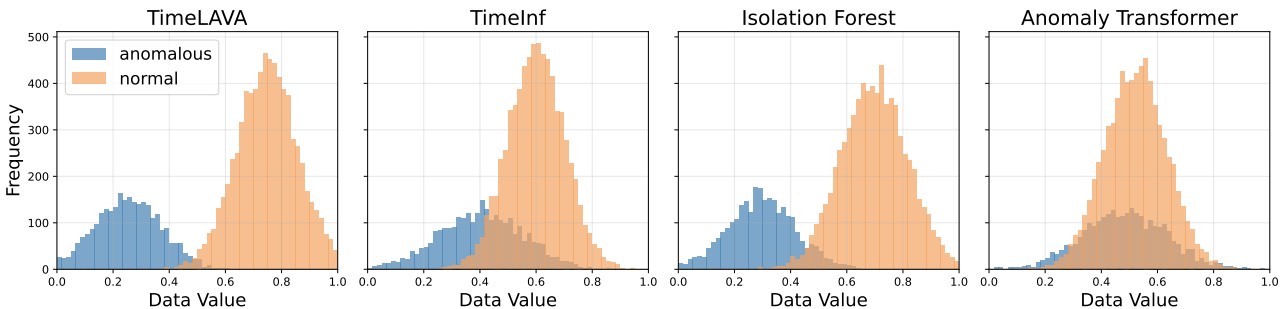

*Figure 9.* **Anomaly score distributions.** TimeLAVA achieves distinct separation between normal and anomalous segments, whereas baselines exhibit significant overlap.

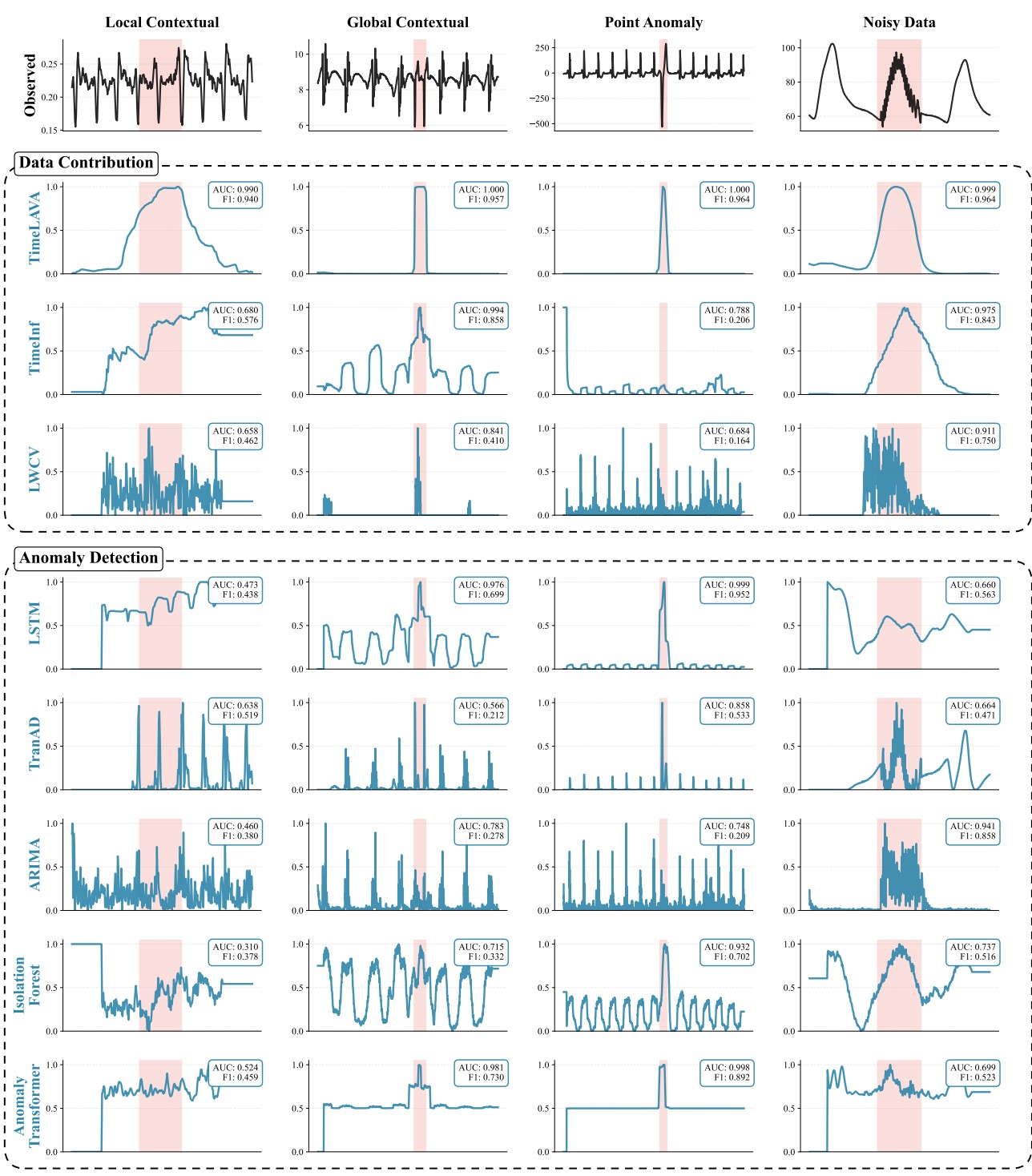

*Figure 10.* Comprehensive results on the **UCR Time Series Anomaly Archive.**

### C.2.2. PARAMETER SENSITIVITY ANALYSIS

In this section, we analyze the impact of key parameters on the performance of TIMELAVA using the SMAP and MSL datasets (Hundman et al., 2018). The main observations can be summarized as follows:

*Table 4.* Supplementary result details. Higher AUC/F1 is better (↑).

| Method | SWaT | | WADI | | NAB-Taxi | | NAB-Tweets | | NAB-AdExchange | | NAB-Traffic | | KDD-Cup99 | |
|---|---|---|---|---|---|---|---|---|---|---|---|---|---|---|
| | AUC ↑ | F1 ↑ | AUC ↑ | F1 ↑ | AUC ↑ | F1 ↑ | AUC ↑ | F1 ↑ | AUC ↑ | F1 ↑ | AUC ↑ | F1 ↑ | AUC ↑ | F1 ↑ |
| Isolation Forest | **0.87** | **0.69** | 0.74 | 0.34 | **0.64** | 0.13 | 0.55 | 0.18 | 0.50 | 0.11 | 0.57 | 0.20 | 0.74 | 0.41 |
| LSTM | 0.30 | 0.16 | 0.49 | 0.08 | 0.33 | 0.04 | 0.55 | 0.17 | 0.56 | 0.12 | 0.57 | 0.19 | **0.98** | 0.88 |
| ARIMA/VAR | 0.46 | 0.11 | 0.41 | 0.04 | 0.49 | 0.05 | 0.56 | 0.17 | 0.51 | 0.14 | 0.55 | 0.20 | 0.87 | 0.52 |
| Anomaly Transformer | 0.60 | 0.22 | 0.50 | 0.06 | 0.51 | 0.06 | 0.50 | 0.08 | 0.44 | 0.07 | 0.46 | 0.05 | 0.54 | 0.17 |
| TimeInf | 0.61 | 0.08 | 0.63 | 0.14 | 0.52 | 0.02 | 0.52 | 0.16 | 0.54 | **0.34** | 0.64 | **0.39** | 0.79 | 0.43 |
| **TIMELAVA (Ours)** | 0.86 | 0.68 | **0.85** | **0.43** | 0.56 | **0.18** | **0.64** | **0.24** | **0.73** | 0.27 | **0.74** | 0.34 | **0.98** | **0.93** |

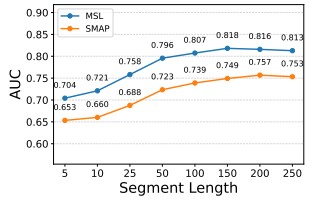 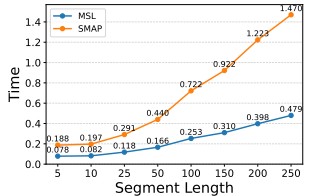

*Figure 11.* Parameter sensitivity analysis on segment length for TIMELAVA.

**Segment Length.** As shown in Fig. 11, increasing the segment length generally leads to improved AUC for both datasets. This observation is consistent with the intuition that longer segments provide a broader temporal context, enabling the wavelet transform to capture more comprehensive patterns and dependencies. For the MSL dataset, AUC improves steadily from $0.704$ at a length of $5$ to approximately $0.818$ at a length of $150$, after which the gains plateau. A similar trend is observed for SMAP. The empirical growth in computational cost observed in Fig. 11 aligns with the theoretical time complexity of the discrete wavelet transform, which is $O(L)$ in the segment length $L$. Hence, there is a clear trade-off between detection performance and runtime: longer segments enhance detection accuracy by capturing richer temporal structure and preserving local patterns, but they also incur higher computational overhead.

Moreover, the appropriate segment length is often domain dependent, as temporal dynamics unfold at different scales in applications such as electricity consumption, mobility, or traffic monitoring.

**Stride.** Fig. 12 illustrates the impact of stride on performance. Smaller stride consistently yields higher AUC for both datasets, as increased overlap between consecutive segments preserves fine-grained temporal information, reducing the risk of missing anomalies of just a few points. As the stride increases from $1$ to $50$, AUC decreases notably, for instance, MSL drops from $0.76$ to $0.68$ and SMAP from $0.69$ to $0.62$. This degradation aligns with the intuition that larger stride will lose certain local information. Therefore, in practice we avoid setting the stride

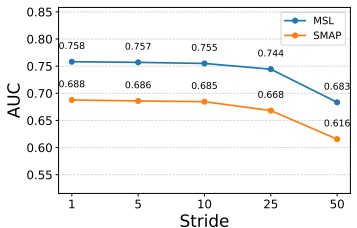

*Figure 12.* Parameter sensitivity analysis on Stride $s$.

too large, as non-overlapping segments risk missing short anomalies and degrading detection accuracy. The observed computational trends agree with the expected complexity that if the sequence length is $T$ and stride is $s$, then the number of

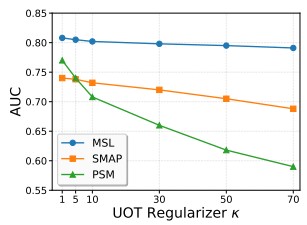

*Figure 13.* Parameter sensitivity analysis on UOT regularizer $\kappa$.

segments is approximately $T/s$, giving an overall cost of $O((T/s) \cdot N)$ (Mallat, 1989). Hence, using $s = 1$ is significantly more expensive than $s = 50$, consistent with the linear dependence on $1/s$.

**UOT Regularizer.** Figure 13 illustrates the effect of the UOT regularization strength $\kappa$ on performance. TIMELAVA remains highly stable for $\kappa$ between 1 and 10, where the AUC varies only minimally; our default choice $\kappa = 2$ lies well within this flat and reliable region. As $\kappa$ increases beyond 30, the formulation begins to resemble balanced OT, which forces mass-conserving matches between segments. This reduces the method's ability to downweight unmatched or outlier segments and leads to a gradual decline in performance. Overall, the trend demonstrates that TIMELAVA is robust to $\kappa$ over a wide operating range, with degradation only at extreme values.

### C.2.3. GENERALITY ANALYSIS

In this section, we explore alternative implementations to the key components of $\mathcal{W}_{\text{SW}}$ to justify its rationale and advantages. Table 5 present an ablation study evaluating the impact of two key components in TIMELAVA: the UOT formulation and the wavelet-based representation.

*Table 5.* **Ablation study on key components of TIMELAVA.** TIMELAVA-OT replaces UOT with OT, and TIMELAVA-Fourier replaces Wavelet with Fourier. **Bold** = best, underline = second best.

| Dataset | TIMELAVA-OT | | | TIMELAVA-Fourier | | | TIMELAVA (Ours) | | |
|---|---|---|---|---|---|---|---|---|---|
| | AUC ↑ | F1 ↑ | Time (s) ↓ | AUC ↑ | F1 ↑ | Time (s) ↓ | AUC ↑ | F1 ↑ | Time (s) ↓ |
| SMD | 0.59 | 0.23 | 24.65 | 0.75 | 0.21 | 24.61 | **0.91** | **0.52** | 14.01 |
| SMAP | 0.74 | 0.48 | 2.13 | **0.79** | **0.63** | 1.04 | 0.74 | 0.54 | 0.72 |
| MSL | 0.79 | 0.49 | 0.68 | **0.84** | **0.57** | 0.41 | 0.81 | 0.49 | 0.25 |
| PSM | 0.59 | 0.38 | 51.18 | 0.74 | 0.57 | 10.68 | **0.77** | **0.58** | 8.16 |
| KDD-Cup99 | **0.98** | 0.92 | 815.46 | **0.98** | 0.92 | 296.01 | **0.98** | **0.93** | 101.52 |
| NAB-Taxi | 0.50 | 0.00 | 0.52 | 0.50 | 0.00 | 0.90 | **0.56** | **0.18** | 0.52 |
| NAB-Tweets | 0.62 | 0.19 | 0.92 | 0.58 | 0.05 | 1.10 | **0.64** | **0.24** | 0.47 |
| NAB-AdEx | 0.71 | 0.31 | 0.05 | **0.76** | **0.47** | 0.09 | 0.73 | 0.27 | 0.02 |
| SWaT | 0.32 | 0.10 | 578.68 | 0.40 | 0.13 | 226.11 | **0.86** | **0.68** | 245.83 |
| WADI | 0.64 | 0.25 | 741.74 | 0.63 | 0.23 | 148.89 | **0.85** | **0.43** | 91.13 |
| NAB-Traffic | 0.60 | 0.21 | 0.09 | 0.48 | 0.00 | 0.04 | **0.74** | **0.34** | 0.03 |

**Effects of the Discrepancy Measure.** Standard OT enforces strict mass-preserving matching constraints, requiring every unit of probability mass to be transported exactly. This becomes problematic under non-stationarity: when anomalies introduce or remove structure, OT spreads the cost across both normal and anomalous regions, blurring detection boundaries. UOT addresses this by relaxing mass conservation through divergence regularization (Fatras et al., 2021; Séjourné et al., 2019), allowing unmatched mass to be penalized directly. This produces sharper, more localized anomaly scores. Replacing UOT with standard OT (TIMELAVA-OT) leads to consistent drops in both AUC and F1 across most datasets (Table 5), with Fig. 14 illustrating how UOT's selective matching yields superior anomaly localization. These empirical findings also corroborate Theorem 5.1, which shows that the distribution patterns of data value scores can distinguish isolated anomalies from systemic regime shifts in real data.

**Effects of the Distance Metric.** Fourier transform provides global frequency decomposition but suffers from poor temporal localization. In contrast, wavelet transform offers joint time-frequency decomposition, preserving both spectral and temporal localization (Fig. 2). This multi-scale representation is crucial for detecting anomalies manifesting as short-lived bursts, shifts, or local contextual changes. Substituting wavelets with Fourier (TIMELAVA-Fourier) generally degrades performance, particularly for localized anomalies. Interestingly, TIMELAVA-Fourier achieves higher scores on SMAP and MSL datasets, but this stems from ground-truth label inaccuracies, where anomalies are annotated as extended intervals rather than precise points (Fig. 15). While TIMELAVA identifies exact anomaly onsets, TIMELAVA-Fourier flags broader regions, better matching the overly wide labels and inflating metrics, though TIMELAVA's finer localization more accurately reflects true anomaly occurrences.

The proposed TIMELAVA, integrating both UOT and wavelet transform, achieves the highest AUC and F1 in the majority of datasets, demonstrating that these design choices contribute synergistically to improved detection accuracy.

### C.3. Data Pruning and Selection

To evaluate the practical effectiveness of TIMELAVA in identifying valuable time series segments, we conduct comprehensive data pruning and selection experiments across multiple public time series datasets. Our experiments compare TIMELAVA against several established baseline methods to show that it achieves the best or second-best performance in data quality assessment.

**Experimental Setup.** We conduct two complementary experiments to assess data quality identification capabilities. In data pruning experiments, we evaluate the ability to identify and remove low-quality segments by progressively eliminating the lowest-valued segments (0% to 50% removal) and measuring downstream forecasting performance, testing whether our method can effectively filter out detrimental data. Conversely, in data selection experiments, we assess data selection capability by retaining only the highest-valued segments (20% to 50% retention) and evaluating performance, measuring the method's ability to identify the most informative data points. For each scenario, we train linear AR models on the selected segments for multi-step forecasting and measure Root Mean Square Error (RMSE), and coefficient of determination ($R^2$). For experiments with injected noise, we assess each method's ability to identify corrupted segments using AUC-ROC. We

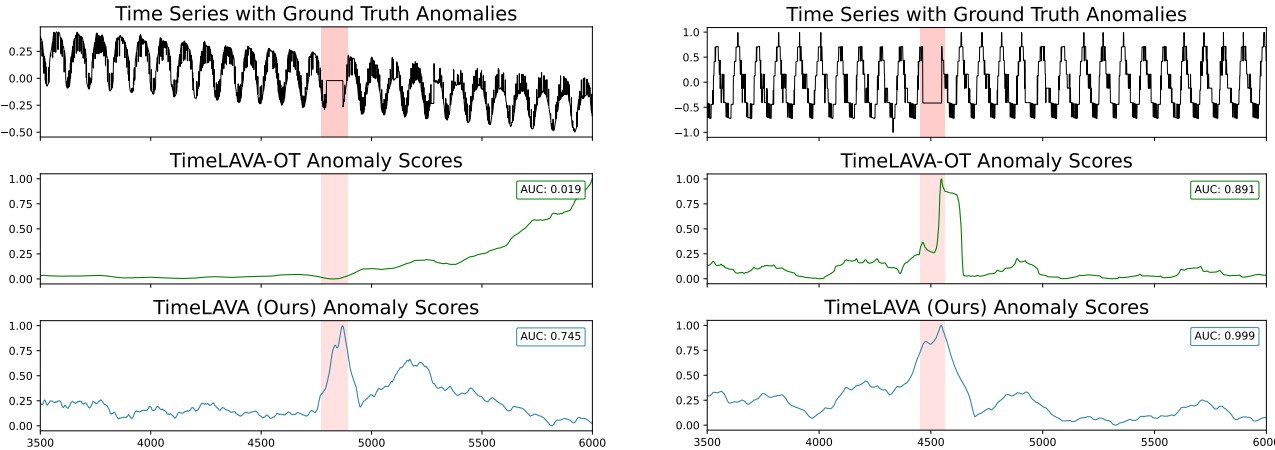

*Figure 14.* Anomaly detection on SMAP channels G-1 (left) and A-2 (right). TIMELAVA with UOT (bottom) shows superior anomaly detection compared to TIMELAVA-OT with standard OT (middle), achieving significantly higher AUC scores.

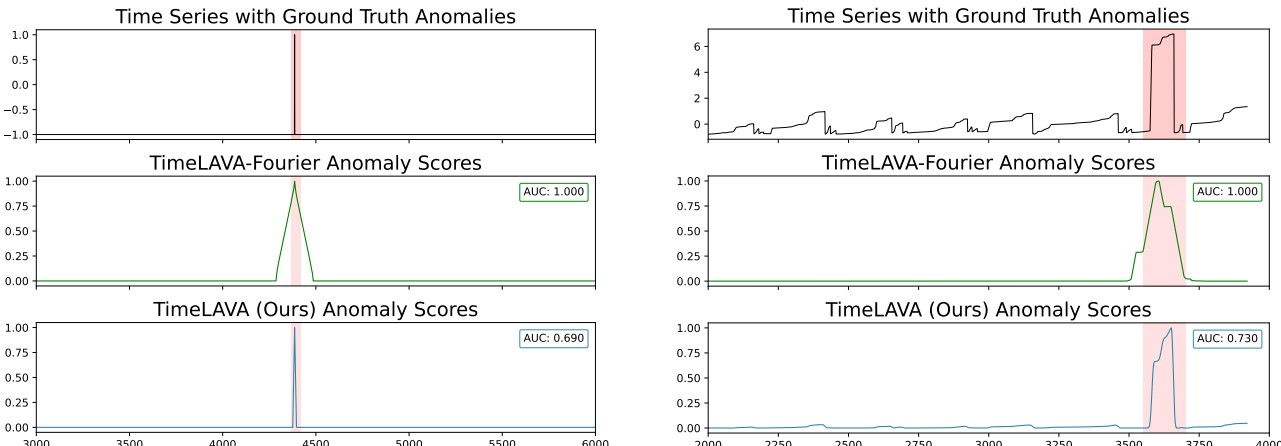

*Figure 15.* Comparison of TIMELAVA-Fourier and TIMELAVA on SMAP channel D-8 (left) and MSL channel F-5 (right). While TIMELAVA-Fourier achieves perfect AUC (1.000), this inflated score results from ground-truth labels that mark extended intervals rather than precise anomaly points. TIMELAVA provides more accurate point-wise localization despite lower AUC.

set $c = 0$ in Eq. 6 since these forecasting tasks focus purely on temporal pattern valuation. For all datasets, we compute valuations per dimension and report the average, restricting inputs to one dimension at a time to avoid excessive runtime for slow methods (e.g., TimeInf) and ensure comparability.

**Datasets.** We evaluate our approach on four widely-used public time series datasets (Wu & Keogh, 2021; Liu et al., 2024).

- **ETT (Electricity Transformer Temperature)**: 2-year hourly data from two counties in China, featuring oil temperature as the target variable and 6 power load features, with the dataset split into 12/4/4 months for train/validation/test.

- **Traffic**: hourly road occupancy rates from 862 sensors across California over 48 months, providing a high-dimensional multivariate time series with complex spatiotemporal patterns.

- **Electricity**: hourly electricity consumption patterns of 321 clients from 2012-2014, representing diverse consumer behavior patterns in energy usage.

- **Exchange**: daily exchange rates for eight countries from 1990-2016, capturing long-term economic trends and currency fluctuations.

For the Electricity, Traffic, and Exchange datasets, we adopt a 70%/10%/20% split for training/validation/testing. We partition each dataset into non-overlapping segments of fixed length, with segment lengths chosen based on dataset characteristics: 96 time steps for ETT and Electricity, 168 for Traffic, and 30 for Exchange. This segmentation allows us to evaluate data quality at a meaningful temporal granularity while maintaining sufficient statistical power for downstream tasks. To test robustness and noise detection capabilities, we systematically corrupt 20% of training segments with four types of realistic time series noise: Gaussian noise (additive white noise scaled by segment standard deviation), spike artifacts (random impulse noise at multiple time points), linear drift (systematic trend distortion across the segment), and scale corruption (multiplicative scaling by random factors).

**Results.** Figs. 16-19 demonstrate TimeLAVA's comparable performance across all datasets.

*Data selection.* In data selection experiments, TimeLAVA consistently achieves the highest $R^2$ values when retaining high-value segments, with particularly strong performance on the Exchange dataset where it maintains high $R^2$ even when using only 30% of the data. The Traffic dataset results highlight TimeLAVA's robustness to high-dimensional, complex temporal patterns, where while baseline methods show significant performance degradation as data is reduced, TimeLAVA maintains stable forecasting accuracy across different retention rates, suggesting that our wavelet-based feature extraction successfully captures essential temporal structures that generalize well to forecasting tasks.

*Data Pruning.* The data pruning results indicate that TimeLAVA effectively removes redundant and low-quality segments without compromising model performance. Fig. 20 presents ROC curves for noise detection across all datasets, where TimeLAVA achieves consistently high AUC scores across different datasets, significantly outperforming baseline methods. The Influence Function method shows competitive performance on some datasets but lacks consistency, while KNN-Shapley and Data-OOB show moderate detection capabilities.

Across all datasets and experimental conditions, TimeLAVA shows consistent performance advantages, maintaining strong performance across diverse temporal patterns, dataset sizes, and noise conditions, suggesting that our approach captures fundamental aspects of time series data quality that generalize across domains.

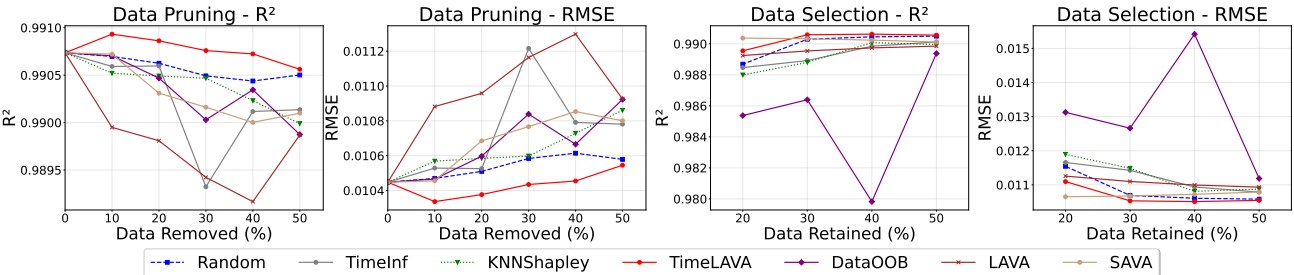

*Figure 16.* Data Selection and Pruning Results - Exchange

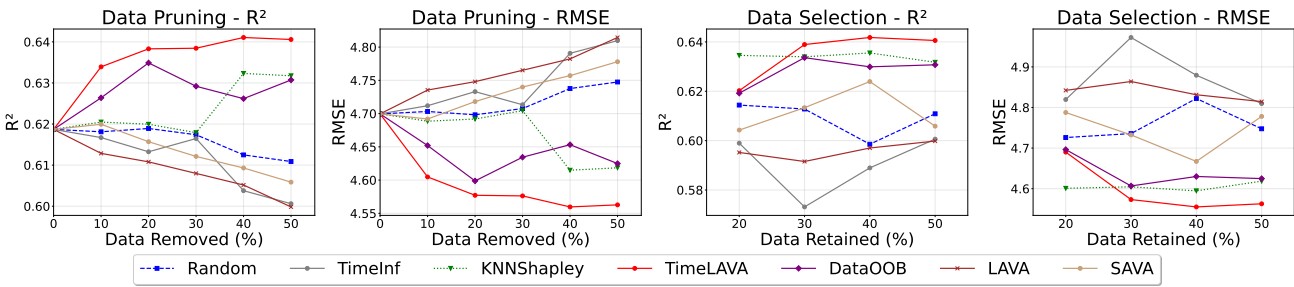

*Figure 17.* Data Selection and Pruning Results - ETTh1

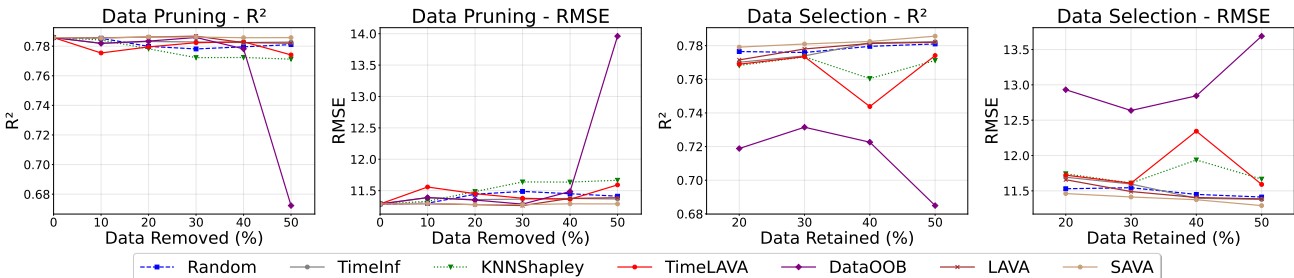

*Figure 18.* Data Selection and Pruning Results - Traffic

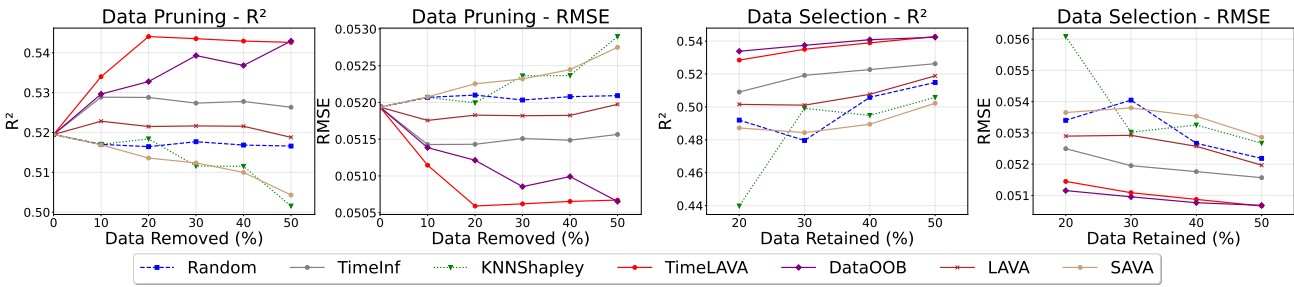

*Figure 19.* Data Selection and Pruning Results - Electricity

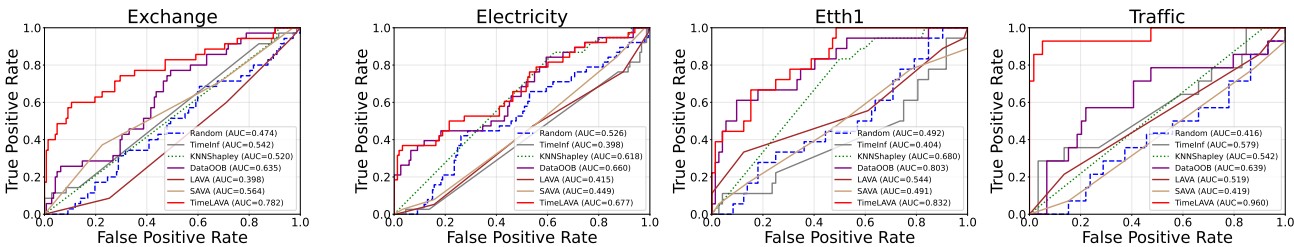

*Figure 20.* **ROC curves** illustrating the performance of temporal label-noise segment identification on four datasets: **Exchange**, **Electricity**, **ETTh1**, and **Traffic**. The results demonstrate the robustness of TIMELAVA in distinguishing clean from corrupted segments across diverse temporal domains.

## C.4. Label Noise

Temporal label noise is a critical challenge in real-world detection systems. Unlike static classification tasks, detection applications often exhibit time-varying labeling quality due to operational factors such as annotator fatigue, sensor drift, or changing environmental conditions. For instance, healthcare monitoring systems show systematic temporal variations: continuous glucose monitors experience calibration drift affecting detection accuracy, while human annotators exhibit time-of-day effects with reduced labeling quality during night shifts (Carpenter, 2003; Hsieh & Kocielnik, 2016; Nagaraj et al., 2025). Traditional approaches assuming static noise distributions fail to capture these temporal dynamics, leading to degraded performance. By explicitly modeling time-varying label quality, our experimental framework enables rigorous evaluation of methods designed to handle temporal label noise versus conventional approaches.

**Experimental Setup.** We consider a controlled semi-synthetic experimental framework that treats original dataset labels as ground truth and systematically injects temporal noise patterns to evaluate detection robustness. This approach follows established benchmarking practices for label noise research (Jiang et al., 2023; Just et al., 2023; Kwon & Zou, 2023) while extending them to capture temporal dynamics essential for detection applications. Since time series classification typically operates on segments rather than individual points, we inject noise at the segment level. After dividing the time series into non-overlapping segments, we corrupt segment labels according to time-dependent probability functions that reflect realistic error patterns observed in operational systems. For each temporal noise pattern, corrupted segments have their labels flipped to the opposite class, simulating common failure modes such as misclassification errors:

$$\tilde{y}_i = \begin{cases} 1 - y_i & \text{if segment } i \in \mathcal{N} \\ y_i & \text{otherwise} \end{cases}$$

where $\mathcal{N}$ denotes the set of segments selected for corruption according to each temporal pattern, and $\tilde{y}_i$ represents the corrupted label for segment $i$. We evaluate four distinct temporal noise patterns (Random, Periodic, Decay, and Growth) that determine when corruption occurs, as detailed below.

Each dataset is partitioned temporally with 70% allocated for training and 30% for validation, ensuring realistic evaluation conditions where future data remains unseen during model development. Temporal noise is injected exclusively into training labels, while validation labels remain clean to provide unbiased performance assessment. This protocol simulates realistic deployment scenarios where detection systems must learn from noisy historical data while being evaluated on clean test conditions. We vary the average noise rate from 5% to 20% to evaluate across different noise severities

**Baselines.** We compare our proposed methods against the same baseline approaches used in our pruning experiments, ensuring consistent evaluation across different aspects of our work.

**Datasets.** We evaluate our temporal detection methods using two real-world datasets from healthcare applications following (Nagaraj et al., 2025). While these datasets were originally designed for standard classification tasks, we adapt them to study temporal label noise by treating the original labels as ground truth and systematically injecting temporal noise patterns. Each dataset represents binary classification tasks over complex temporal feature spaces that are well-suited for evaluating robustness to time-varying label corruption. The datasets include:

- **Moving**: human activity recognition task where we detect movement states (e.g., walking vs. sitting) using temporal accelerometer data in adults (Reyes-Ortiz et al., 2013).

- **Senior**: similar human activity recognition task as above but in senior citizens (Logacjov & Ustad, 2023).

- **Blinking**: eye movement (open vs closed) detection task using continuous EEG data (Roesler, 2013).

These classification tasks provide suitable testbeds for temporal label noise evaluation because they involve sequential labeling over time where temporal corruption patterns can realistically occur in practical deployment scenarios.

**Temporal Noise Injection.** Our noise injection mechanism corrupts detection labels according to time-dependent probability functions that reflect realistic error patterns observed in operational detection systems. We evaluate four distinct temporal noise patterns that capture diverse real-world detection error scenarios:

- **Random Noise** provides a baseline with uniform corruption probability: $P(i \in \mathcal{N}) = \eta$ for all segments $i$, representing temporally independent errors.

- **Periodic Noise** models cyclical error patterns: $P(i \in \mathcal{N}) = \frac{\eta}{2} + \frac{\eta}{2}\sin(\omega t_i + \phi)$ where $t_i$ denote the temporal position of segment $i$, $\omega$ controls frequency and $\phi$ is the phase offset, capturing systems affected by daily cycles or periodic maintenance.

- **Decay Noise** represents exponentially decreasing error rates: $P(i \in \mathcal{N}) = \eta \cdot \exp(-\lambda t_i/T)/Z$ where $\lambda$ controls decay rate and $Z$ is a normalization constant ensuring overall noise rate $\eta$. This models systems improving through learning or calibration.

- **Growth Noise** captures deteriorating performance: $P(i \in \mathcal{N}) = \eta/(1 + \exp(-\beta(t_i - T/2)))/Z$ where $\beta$ controls growth rate, $T/2$ is the midpoint, and $Z$ normalizes to achieve target rate $\eta$. This reflects aging or drift effects.

As shown in Fig. 21, these temporal noise patterns create distinct corruption patterns in the actual label sequences, with corrupted labels distributed according to their respective temporal probability functions.

**Evaluation.** We evaluate noise detection capability using F1-score metrics to assess how effectively each method identifies temporally corrupted detection labels.

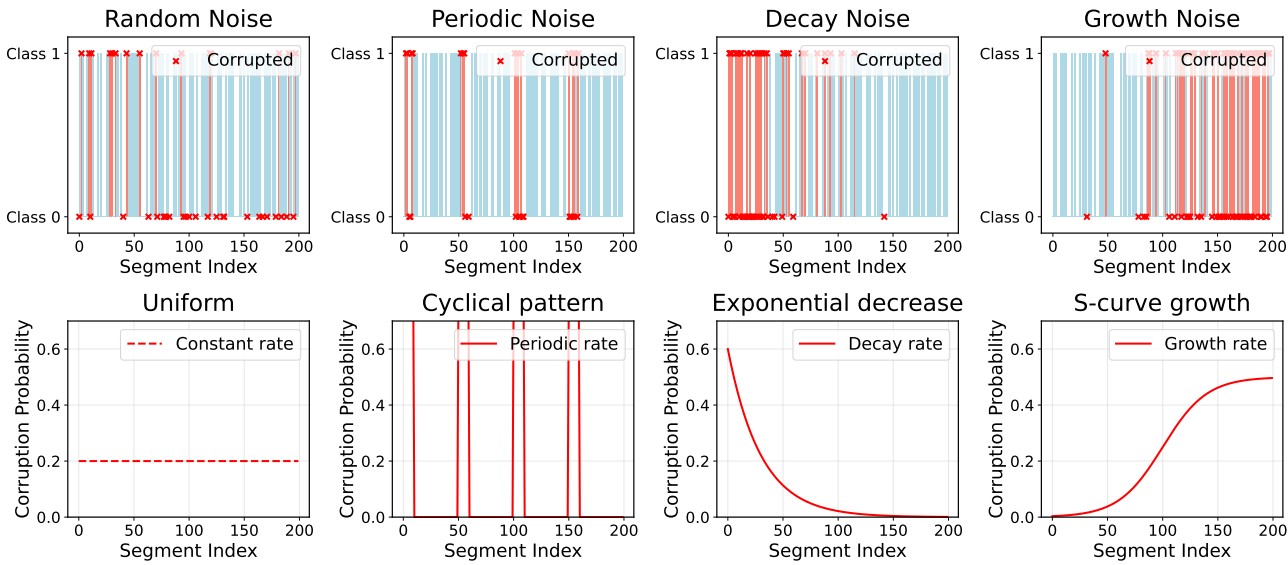

*Figure 21.* **Temporal label noise patterns in time series segments.** Upper: Binary classification labels (Class 0/1) with corrupted segments marked in red. Lower: Time-varying corruption probability $P(i \in \mathcal{N})$ for each pattern. Random noise maintains constant probability $\eta$, Periodic follows sinusoidal variation simulating cyclical quality changes, Decay shows exponential decrease modeling learning effects, and Growth exhibits sigmoid increase representing degradation over time.

**Results.** As shown in Figs. 22–24, TIMELAVA achieved the best performance in periodic noise detection, substantially outperforming all baseline methods, while demonstrating competitive performance on other noise patterns. KNN-Shapley performs well as it assumes that segments with similar features should have similar labels, making it effective at detecting mislabeled segments that disagree with their nearest neighbors in the feature space when label noise percentage is large. TIMELAVA's advantage stems from its wavelet-based features that capture multi-scale temporal patterns within each segment, making it more sensitive to systematic labeling errors. Unlike methods that treat time series segments as independent feature vectors, TIMELAVA preserves the sequential nature of temporal data through its sliding window approach and distributional alignment, making it effective for real-world scenarios where labeling errors follow temporal patterns such as periodic system failures, time-dependent annotation quality degradation, or systematic errors that evolve over time.

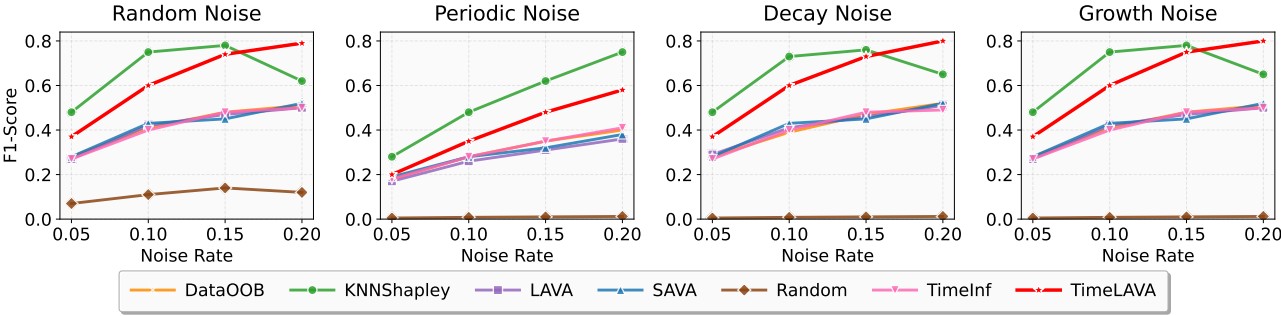

*Figure 22.* F1-score on **Blinking** across different noise types.

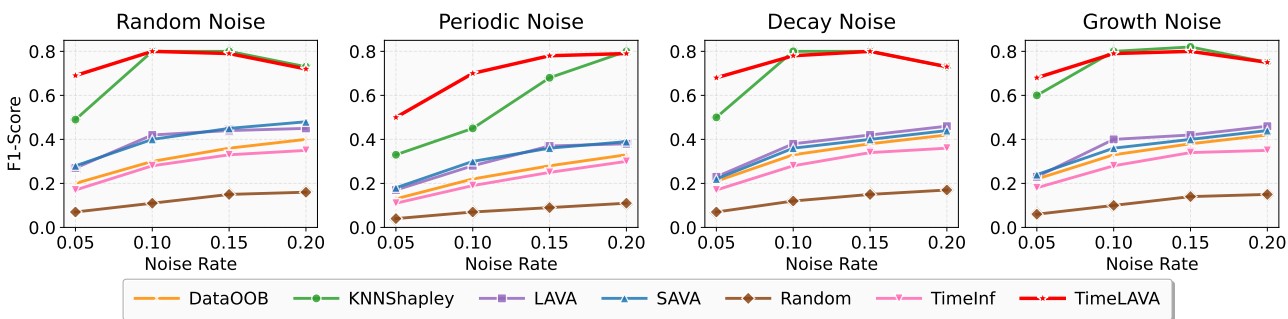

*Figure 23.* F1-score on **Moving** across different noise types.

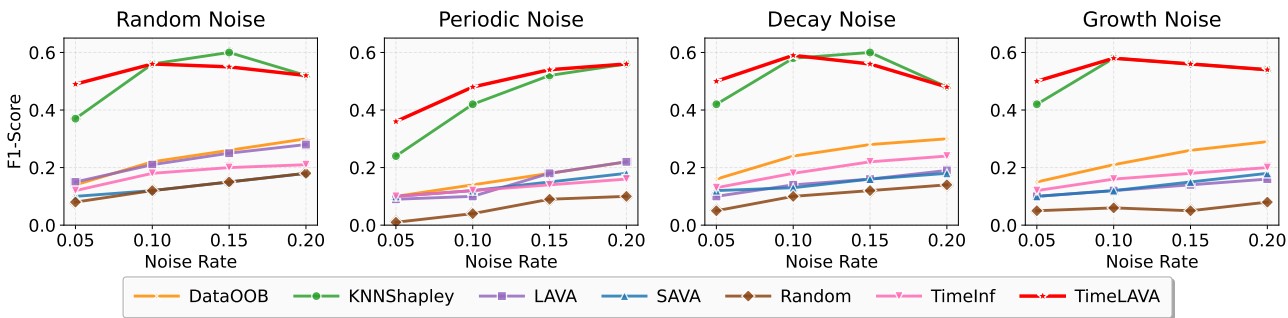

*Figure 24.* F1-score on **Senior** across different noise types.

**Parameter Sensitivity Analysis.** We conduct sensitivity analyses for the two core hyperparameters: the UOT regularization strength $\kappa$ and the label balance parameter $c$.

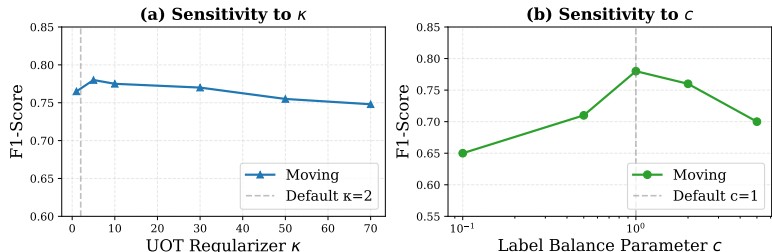

*Figure 25.* Sensitivity to hyperparameters $\kappa$ and $c$ on Moving dataset.

**UOT Regularizer $\kappa$.** Figure 25 (a) shows the effect of UOT regularization strength $\kappa$ on label noise detection (Moving dataset, 15% periodic noise, $c = 1$). TIMELAVA demonstrates remarkable stability for $\kappa \in [1, 10]$, with F1-score varying less than 1.3% (0.765–0.775); our default $\kappa = 2$ lies well within this stable region. As $\kappa$ exceeds 30, the formulation resembles balanced OT, enforcing strict mass conservation that reduces the method's ability to downweight outliers, leading to performance decline (F1=0.700 at $\kappa = 70$). TIMELAVA is thus robust to $\kappa$ over a wide range, with degradation only at extreme values.

**Label Balance Parameter $c$.** Figure 25 (b) shows the effect of label balance parameter $c$ (fixing $\kappa = 2$). Performance exhibits an inverted-U pattern, peaking at $c = 1$ (F1=0.78). When $c < 1$, the model underweights label information; at $c = 0.1$, F1 drops to 0.65 as the method behaves like an unsupervised matcher. When $c > 1$, the model overemphasizes label alignment; at $c = 5$, F1 degrades to 0.70 as it becomes less robust to noisy labels. The stability around $c \in [0.5, 2.0]$ (F1: 0.71–0.78) validates our default choice $c = 1$, achieving robust balance between temporal patterns and label consistency.

