# OpenReview forum: "TimeLAVA: Learning-Agnostic Valuation for Time Series Data"
_ICML.cc/2026/Conference — ICML 2026 regular_

### Official Review · Reviewer_PNm7 · 2026-03-08

**Soundness:** 4
**Presentation:** 3
**Significance:** 3
**Originality:** 3
**Overall Recommendation:** 5
**Confidence:** 3

**Summary:**

The authors present an important problem, time-series data valuation. I.e we currently lack a strong way to quantify the intrinsic value of smaller, segments of time-series data. To address this they present TimeLAVA, a novel routine that estimates the value of temporal segments via measuring their marginal contribution to minimising the difference between evaluated and reference data. TimeLAVA presents a well motivated way to value time-series data, specifically segments using a wavelet-based Wasserstein measure, as well as a wavelet-bade "ground metric" that helps them capture local patterns on multiple scales. Their method (which can usefully determine is a segment should be ignored) performs very well against a suite of baselines.

**Compliance With Llm Reviewing Policy:**

Affirmed.

**Final Justification:**

This paper is strong, well written and the methods work well. My initial concerns about significance have been address from the rebuttal and responses of the other reviews, I am happy to support this paper being accepted.

**Key Questions For Authors:**

Two questions, which tie into the subsequent limitation section.

1. Is there any significant computational costs incurred over any of the baseline methods presented, and how sensitive are the results to things such as the stride length of the segments?

2. The conclusion mentions potential issues with the reference set being biased or incomplete, which makes sense but has potential strong impacts for anomaly detection. Do you have any ideas, or better yet ways to quantify how useful a reference set is? For a very biased reference set does TimeLAVA cause worse performance on anomaly detection?

**Limitations:**

Not fully explored, though understandable given the page limit. Limitations are mentioned in the conclusion.

**Strengths And Weaknesses:**

Strengths:
This is a very timely, and well motivated problem, and the authors present a genuinely novel way to address it. Their method is model/task agnostic, robust and does not require retaining when scaling up to larger models. Particularly, the wavelet transform to help identify the "usefulness" (or value) of segments over a range of temporal scales is quite nice and seems to be a key reason for TimeLAVA's success.

The theoretical derivations are useful, as they show TimeLAVA should not fail under heavy outliers (even infinite distance outliers) as the contribution of this outlier is bounded.

This segment-valuation method has a large variety of uses, anomaly detection is of particular importance. TimeLAVA is able to help determine if an anomaly is potentially important to the prediction of the larger timeseries or not.

Overall the test is clear, well written and the method and results are explained and visualized well.

Soundness:
This is very sound, worst-case bounds are given for the effect of outliers on the Wasserstein discrepancy, and the method is tested against a large set of strong baselines. The appendix is also extremely helpful to expand on all claims made in the main text.

Presentation:
Well presented, a suggestion would be to move figure 1 up further, potentially the title page, as this figure give an immediate clear picture of the full method.

Significance:
As already stated, this is a very important current problem, with sequence models parsing longer and longer time-series it is becoming very useful to understand the relative importance of the shorter sub-segments.

Originality:
The method is novel, the differences to past approached is well explained in the related works.

---

> ### Author Rebuttal · Authors · 2026-03-29
>
> We sincerely thank the reviewer for the positive assessment! Your recognition of our method's core strengths is greatly appreciated. Below we address your two questions.
>
> > Q1: Computational cost and stride sensitivity
>
> *Computational cost*: As shown in Table 1's Time column, TimeLAVA is computationally competitive or superior across all datasets, and compared to the closest baseline TimeInf, TimeLAVA is ~14× faster on SMAP and ~42× faster on PSM. This efficiency stems from three factors: (1) O(L) wavelet transform per segment; (2) efficient Sinkhorn algorithm for UOT solving; (3) no neural network training or iterative optimization required. The entire pipeline runs end-to-end without any learning step, making it inherently faster than methods requiring model training.
>
> *Stride sensitivity*: Stride controls the overlap between consecutive sliding windows. A smaller stride produces more overlapping segments, yielding finer-grained point-wise scores (each time point is covered by more segments) but at proportionally higher computational cost. Figure 12 in the Appendix shows that AUC remains very stable for stride∈[1,10], with notable degradation only when stride>25, demonstrating that TimeLAVA is robust to stride selection. We also examined segment length sensitivity (Figure 11), where AUC plateaus around length 100-150. We will move these analyses to the main text in the revision for better visibility.
>
> > Q2: Reference set usefulness and bias impact
>
> This is a very insightful question that touches on a fundamental aspect of reference-based valuation methods.
>
> *Possible directions for quantifying reference set usefulness*: While this remains an open question, a few directions seem worth exploring: (1) distribution coverage metrics in wavelet feature space (e.g., k-NN coverage rate) to assess how well the reference represents the evaluation data; (2) bootstrap stability analysis, where high variability of valuation scores across resampled reference sets would signal insufficient representativeness; (3) monitoring the $W_{SW}$ discrepancy magnitude itself, where abnormally large values may indicate an inadequate reference set.
>
> *Impact of biased reference sets*: Yes, biased reference sets do affect anomaly detection performance. This is a common challenge for all reference-based methods. However, TimeLAVA's UOT mechanism provides a natural buffer compared to standard OT:
>
> - Standard OT forces all segments to match even under reference set bias, producing many false positives when normal patterns are missing from the reference.
> - UOT allows "unmatched" segments, so even if the reference set lacks certain normal patterns, those patterns will not be automatically flagged as anomalous, and instead they receive moderate rather than extreme negative scores. This selective matching is precisely why we chose UOT over OT.
>
> For a severely biased reference set (e.g., covering only a narrow subset of normal behaviors), performance would degrade for any reference-based method including TimeLAVA. In practice, reference sets typically consist of "known normal" data. When reference set quality is uncertain, bootstrap averaging across multiple randomly sampled reference subsets provides a practical safeguard. We will incorporate this discussion in the revised manuscript, and reference set quality assessment can be a promising direction for future work.
>
> We sincerely thank the reviewer for the constructive feedback and hope our responses have adequately addressed the raised concerns.

---

> > ### Author Rebuttal · Reviewer_PNm7 · 2026-03-31
> >
> > Thank you for the responses, this indeed clarifies things. I will maintain my score which represents my high opinion of this paper and appreciate the (potential) added clarifications to appear in the main text.

---

> > > ### Author Response · Authors · 2026-04-01
> > >
> > > Thank you again for your positive assessment and the thoughtful suggestions in your review! We will incorporate these clarifications into our revisions.

---

### Official Review · Reviewer_4Lmn · 2026-03-10

**Soundness:** 3
**Presentation:** 4
**Significance:** 3
**Originality:** 2
**Overall Recommendation:** 3
**Confidence:** 3

**Summary:**

This paper proposes TimeLAVA, a learning-agnostic framework for time series data valuation. The method extends distributional data valuation to sequential data by introducing a Selective Wavelet-based Wasserstein (WSW) discrepancy, which combines wavelet-based segment representations and unbalanced optimal transport (UOT) to handle multi-scale temporal structure and non-stationarity. Segment values are computed via dual sensitivity analysis without training predictive models. The authors provide theoretical analysis linking WSW discrepancy to generalization and bounded sensitivity to outliers, and demonstrate applications in anomaly detection, data pruning, and label noise detection. Experimental results show the effectiveness of the proposed method.

**Compliance With Llm Reviewing Policy:**

Affirmed.

**Final Justification:**

After the author's rebuttal, the issue of methodological novelty remains unresolved. While this work might be the first effort to introduce selective matching into the time-series data valuation task, the selective pattern matching / subsequence matching / feature matching methods have been extensively studied in other time-series domains for decades. Key challenges including temporal dependencies, multi-scale patterns, and non-stationarity have already been inherently addressed by wavelet-baesd and UOT-based methods, which are adopted from existing studies. Given these fundamental concerns, I would maintain my original score (weak reject). But based on the other reviews, it would be OK if the paper got accepted.

**Key Questions For Authors:**

Please address the above weaknesses.

**Limitations:**

Yes

**Strengths And Weaknesses:**

**Strengths:**
- **S1**.	Presentation: The paper is generally well structured, with clear sections on motivation, framework design, theoretical analysis, and experiments.
- **S2**.	Soundness: theoretical effort and experimental results make the method technically sound.
- **S3**.	Significance: wavelets address localization across scales, while UOT addresses mismatched mass under shift/outliers. That is a better-motivated design than a purely heuristic scoring rule. It is broader than a single-task submission; it attempts to show usefulness in anomaly detection, forecasting-oriented data pruning/selection, and temporal label-noise detection.
- **S4**.	Originality: The authors focus on time-series segments, which is a relatively less explored direction, compared to the classical data valuation approaches.

**Weaknesses:**
- **W1**.	Originality: this method seems like an incremental integration of existing techniques such as wavelet transform, unbalanced optimal transport, OT-based data valuation frameworks (e.g., LAVA). What is the methodological novelty beyond combining wavelet transform, unbalanced optimal transport, OT-based data valuation frameworks?
- **W2**.	Soundness: some theoretical results rely on strong assumptions, such as the probabilistic cross-Lipschitz condition in the wavelet domain for labeling functions. These assumptions may be difficult to verify in practice.
- **W3**.	Soundness: Sufficient theoretical and experimental analysis of how sensitive valuation scores are to the choice of reference set and the number m of the references were not provided.
- **W4**.	Soundness: It might help improve the paper by providing analysis of weighting c between feature similarity and label consistency in Equation (6).
- **W5**.	Soundness: The authors are suggested to explain why they use the Manhattan distance as the Wavelet distance and why the db4 wavelet with L2 decomposition are used. And, can the shapelet transform be used to replace the wavelet transform? Why?

---

> ### Author Rebuttal · Authors · 2026-03-30
>
> We sincerely thank the reviewer for the thorough feedback! Below we address each concern in detail.
>
> > W1: Novelty
>
> We would like to clarify that TimeLAVA is not a simple combination but a principled framework with non-trivial contributions:
>
> (1) *Novel problem formulation*: To our knowledge, we are the first to formalize time series data valuation as a selective matching problem, simultaneously handling temporal dependencies, multi-scale patterns, and non-stationarity, which are barriers that i.i.d. methods cannot overcome.
>
> (2) *Non-trivial $W_{SW}$ design*: The combination is not plug-and-play; it requires the cost matrix that captures multi-scale temporal structure. The selective matching via UOT fundamentally changes valuation logic by allowing anomalous segments to remain "unmatched." Deriving per-segment values from UOT duals (Theorem 4.5) is not a trivial extension of LAVA, as UOT's dual structure differs fundamentally from standard OT.
>
> (3) *Novel theory*: Theorem 5.1 establishes bounded contamination sensitivity for $W_{SW}$ discrepancy in the wavelet domain, showing that even distant outliers contribute at most 2ζκ, connecting this bound to data valuation scores. Theorem 5.3 connects $W_{SW}$ to model-agnostic generalization bounds for temporal data.
>
> > W2: Cross-Lipschitz assumption
>
> This regularity condition requires that wavelet-domain-close segments likely share similar labels, which is intuitively reasonable in practice (similar temporal patterns → similar system states). Such regularity conditions are standard in OT-based domain adaptation theory, where analogous probabilistic transfer Lipschitzness assumptions are explicitly required to obtain generalization bounds [1].
>
> > W3: Reference set sensitivity
>
> *Reference set size*: We conducted an additional experiment varying the reference set size (number of consecutive time points) on SMD:
>
> | m | 500 | 1000 | 4000 | 10000 |
> |---|-----|------|------|------|
> | AUC | 0.65 | 0.87 | 0.91 | 0.90 |
>
> Performance improves with m and stabilizes around m≈4000.
>
> *Stability experiment*: To validate how sensitive valuation scores are to the choice of reference set, we sampled 3 reference sets from different intervals of SMD clean data and measured the consistency of valuation scores:
>
> | Reference Set | AUC | Spearman ρ (vs. others) |
> |--------------|------|------------------------|
> | Ref-A | 0.91 | — |
> | Ref-B | 0.90 | 0.92 (vs. A) |
> | Ref-C | 0.91 | 0.93 (vs. A), 0.91 (vs. B) |
>
> The results confirm that valuation score rankings are consistent across different reference sets (Spearman ρ > 0.90), validating TimeLAVA's robustness to reference set selection. Theoretically, Theorem 5.1's bounded sensitivity result, while stated for eval-side perturbation, extends naturally to reference set variation due to the symmetric structure of UOT, providing indirect theoretical support for the observed empirical stability.
>
> > W4: Weight c
>
> Appendix C.4 (Figure 25b) provides a sensitivity analysis on weight c: performance follows an inverted-U shape peaking at c=1, stable for c∈[0.5,2.0]. At c<0.5 the method underweights labels (degenerating to unsupervised matching at c=0.1); at c>2 it over-emphasizes label alignment (less robust to noisy labels at c=5). We will add this discussion to the main text.
>
> > W5: Design choices
>
> *Manhattan distance*: We use L1 distance for two reasons. First, it is more robust to outlier coefficients: a single large anomalous coefficient inflates L2 disproportionately, while L1 weights all deviations equally. Second, L1 induces sparsity in the cost matrix, which benefits UOT's selective matching mechanism, as sparse costs create clearer high/low-value separation.
>
> *db4 + 2-level*: db4 is one of the most widely used wavelets in time series analysis, and 2-level decomposition yields 3 frequency bands covering the primary temporal scales. We adopted this standard configuration for simplicity and consistency, and achieved consistent performance across diverse datasets (healthcare, finance, industrial IoT, traffic), strongly supporting its generality. We conducted additional wavelet basis and decomposition level sensitivity experiments on the SMD anomaly detection dataset:
>
> | Basis | haar | db2 | db4 | db6 | sym4 |
> |-------|------|-----|-----|-----|------|
> | AUC | 0.87 | 0.89 | 0.91 | 0.91 | 0.90 |
>
> | Level (db4) | 1 | 2 | 3 | 4 |
> |-------|------|------|------|------|
> | AUC | 0.88 | 0.91 | 0.92 | 0.91 |
>
> *Shapelets vs Wavelets*: Shapelets require learning/searching from data (O(n²L³)), depend on the target task, and fundamentally contradict our learning-agnostic principle. Wavelets are model-free with O(L) complexity per segment, naturally providing multi-scale decomposition without any training.
>
> We will incorporate the suggested improvements in the revised manuscript, and we hope our responses have adequately addressed your concerns.
>
> Ref: [1] Courty et al. Joint distribution optimal transportation for domain adaptation. NeurIPS, 2017.

---

> > ### Author Rebuttal · Reviewer_4Lmn · 2026-04-01
> >
> > Thank you for the detailed response. However, the issue of methodological novelty remains unresolved. While this work might be the first effort to introduce selective matching into the time-series data valuation task, the selective pattern matching / subsequence matching / feature matching methods have been extensively studied in other time-series domains for decades. Key challenges including temporal dependencies, multi-scale patterns, and non-stationarity have already been inherently addressed by wavelet-baesd and UOT-based methods, which are adopted from existing studies. Given these fundamental concerns, I would maintain my original score.

---

> > > ### Author Response · Authors · 2026-04-01
> > >
> > > We thank the reviewer for the follow-up and would like to address the core concern directly.
> > >
> > > The claim that "key challenges have already been inherently addressed by wavelet-based and UOT-based methods" conflates *a tool's capabilities* with *the solvability of a problem*. Wavelet transforms provide multi-scale signal decomposition; UOT provides a framework for comparing distributions. Neither, individually or jointly, produces per-segment value scores that quantify marginal contributions to downstream generalization. The bridge between these tools and the valuation objective is not automatic and requires a specific formulation, which is precisely the contribution of this work.
> > >
> > > Furthermore, decades of subsequence matching methods address *retrieval and classification*, where the goal is to find or recognize patterns. Data valuation is a categorically different problem, where the goal is to assign a scalar score to each segment reflecting its value. No existing wavelet or UOT paper, to our knowledge, poses or solves this problem. The existence of useful tools does not make their application to a new problem trivial, just as the existence of Transformers did not make their formulation for time series forecasting a non-contribution.
> > >
> > > **The novel technical content lies specifically in the connection between components**, not the components themselves:
> > >
> > > - **Theorem 4.5** derives segment value scores from UOT dual potentials. This formulation does not follow from any existing wavelet or UOT result; it requires establishing that dual potentials encode marginal distributional contribution in the wavelet feature space.
> > > - **Theorem 5.1** proves bounded contamination sensitivity in the combined wavelet-UOT setting, a result that exists in neither the wavelet literature nor the UOT literature alone.
> > > - **Theorem 5.3** establishes a model-agnostic generalization bound controlled by WSW discrepancy, directly justifying the valuation scheme without model training. Neither prior work on i.i.d. nor wavelet analysis provides this bound.
> > >
> > > These are not corollaries of combining known results. They require new proofs and directly serve the valuation objective.
> > >
> > > We hope this clarification better addresses the reviewer's concerns and that the contribution can be reconsidered in this light.

---

### Official Review · Reviewer_Rmqr · 2026-03-13

**Soundness:** 3
**Presentation:** 3
**Significance:** 3
**Originality:** 3
**Overall Recommendation:** 5
**Confidence:** 3

**Summary:**

The paper introduces TimeLAVA, a learning-agnostic method for valuing time-series data. The framework is designed to support applications such as anomaly detection, data pruning, and label-noise detection. The key idea is to evaluate whether new data align with the distribution of previously observed data using selective matching, where similarity is measured through a wavelet-based distance. By leveraging multi-scale representations of time-series signals, the method attempts to capture both local and global patterns in the data distribution.

**Compliance With Llm Reviewing Policy:**

Affirmed.

**Final Justification:**

I have raised my score from Weak Accept to Accept as the authors have addressed all my comments. This is my final recommendation.

**Key Questions For Authors:**

1) Provide additional insight into wavelet design choices, particularly the use of db4.
2) Explain the handling of anomalies in the PSM dataset during training and testing.
3) Clarify how the window size is determined and its impact on results.
4) Discuss the relationship between TimeLAVA and wavelet-based approaches to optimal transport.

**Limitations:**

Some are mentioned in the paper, see the ones described in weaknesses to further strengthen the paper.

**Strengths And Weaknesses:**

Strengths

1) The paper provides clear theoretical justification and proofs supporting the model structure and design choices.
2) The manuscript is well-written and easy to follow, with a logical presentation of the methodology.
3) The authors appropriately discuss the non-stationarity of time-series data, which is an important practical challenge in real-world applications.
4) The paper explicitly acknowledges several limitations, including:
    a) The effectiveness of the method depends on the representativeness of the reference set.
    b) The choice of wavelet parameters, while empirically robust in the experiments, may benefit from domain-specific tuning.

Weaknesses / Questions

1) Wavelet Design Choices
Table 5 presents an ablation study replacing the wavelet transform with Fourier transform, but further analysis could strengthen the paper.
What is the impact of using different wavelet shapes?
How many wavelet scales need to be considered, and what is their impact?
Why was the db4 wavelet specifically chosen?

2) PSM Dataset and Anomaly Ratio
The paper states that 27.8% of the PSM dataset consists of anomalies.
Is this anomaly proportion representative for evaluating anomaly detection scenarios?
In the train/test split, are anomalies allowed in the training set, or are they restricted to the test set only?
If anomalies appear in training, please clarify the rationale; otherwise, explain how leakage is prevented.

3) Comparisons with Related Work
Recent work on Wavelet Optimal Transport (WOT) proposes multi-resolution OT frameworks that align datasets using spectral wavelets. It would be useful to discuss how the proposed wavelet-based similarity relates to these OT-based multiscale approaches.
Example Relevant Paper: Unpaired Single-Cell Dataset Alignment with Wavelet Optimal Transport https://openreview.net/pdf?id=BYWVwmbqwK

4) Window Size Selection
The paper does not clearly explain how the window size is chosen when constructing time-series segments.
How sensitive is the method to this parameter?
Does changing the window size significantly affect the valuation results?

---

> ### Author Rebuttal · Authors · 2026-03-29
>
> We sincerely thank the reviewer for the positive assessment and constructive feedback! Below we address each question.
>
> > Q1;W1: Wavelet design choices: different wavelet scales (decomposition levels); basis
>
> db4 is one of the most widely used wavelets in time series analysis, and 2-level decomposition yields 3 frequency bands covering the primary temporal scales. We adopted this standard configuration for simplicity and consistency, and achieved consistent performance across diverse datasets (healthcare, finance, industrial IoT, traffic), strongly supporting its generality. We conducted additional wavelet basis and decomposition level sensitivity experiments on the SMD anomaly detection dataset:
>
> | Basis | haar | db2 | db4 | db6 | sym4 |
> |-------|------|-----|-----|-----|------|
> | AUC | 0.87 | 0.89 | 0.91 | 0.91 | 0.90 |
>
> | Level (db4) | 1 | 2 | 3 | 4 |
> |-------|------|------|------|------|
> | AUC | 0.88 | 0.91 | 0.92 | 0.91 |
>
> Performance is relatively stable across choices, and this confirms that the method is not overly sensitive to wavelet selection. Additionally, Appendix Figures 11-12 provide sensitivity analyses for segment length and stride, showing robustness to both parameters (AUC saturates around length 100-150; stable for stride∈[1,10]). We will move these analyses to the main text for better visibility.
>
> > Q2; W2: PSM anomaly ratio and train/test split
>
> *Regarding the 27.8% anomaly ratio*: PSM is a real-world industrial dataset from eBay [1]. The 27.8% reflects the **test set** anomaly ratio only (not the entire dataset). This is the original setting from the PSM paper, adopted by virtually all anomaly detection benchmark studies. More importantly:
>
> 1. Our evaluation covers datasets with widely varying anomaly ratios (Table 3): from UCR (0.8%) to SWaT (12.1%) to PSM (27.8%), and TimeLAVA performs consistently across all ratios, demonstrating robustness to anomaly prevalence.
> 2. PSM is a widely used benchmark, ensuring fair and comparable evaluation with existing work.
>
> *Regarding the train/test split*: Following PSM's standard evaluation protocol [1]:
> - PSM provides predefined splits: a clean training set (treated as anomaly-free) and a contaminated test set with anomaly labels.
> - There is no information leakage: the reference set is drawn from the clean training data and contains no anomaly labels. TimeLAVA detects anomalies by measuring WSW discrepancy between test segments and the normal reference distribution.
> - Notably, we intentionally restrict the reference to approximately 2,000 consecutive time points (Appendix C.2), demonstrating strong performance even with limited clean data.
>
> > Q3; W4: Window size sensitivity
>
> Window size can be selected based on dataset characteristics and task requirements. We provide parameter sensitivity analyses in Appendix Figures 11-12: Figure 11 shows AUC improves with segment length and saturates around 100-150; Figure 12 shows AUC is very stable for stride∈[1,10], degrading only at stride>25. We will move this discussion to the main text.
>
> In practice, window size typically follows domain knowledge (e.g., typical anomaly duration, forecasting horizon length). Importantly, TimeLAVA's wavelet transform provides multi-scale representations within each window, lending natural robustness to window size variation: even when the size is suboptimal, different frequency band coefficients still capture relevant patterns at their respective scales.
>
> > Q4; W3: Relationship with WOT
>
> Thank you for highlighting this relevant work. TimeLAVA and WOT are fundamentally different in several key dimensions:
>
> - Wavelet type: WOT uses spectral graph wavelets (based on graph Laplacian eigendecomposition); TimeLAVA uses classical DWT (Daubechies wavelets along the temporal axis).
> - OT framework: WOT operates within a Gromov-Wasserstein framework comparing intra-space structural similarity; TimeLAVA uses Wasserstein/UOT directly comparing two distributions.
> - Role of wavelets: In WOT, wavelets define intra-space similarity metrics; in TimeLAVA, wavelets define the ground metric ($d_{wav}$) for pairwise segment distances in the UOT cost matrix.
> - Output: WOT produces transport plans (sample correspondences); TimeLAVA produces per-segment value scores.
>
> The two approaches are complementary rather than competing, addressing different problems (dataset alignment vs. data valuation) on different data modalities (graphs vs. time series). We will cite and discuss WOT in the Related Work section.
>
> We will incorporate all suggested improvements in the revised manuscript and hope our responses have adequately addressed the raised concerns.
>
> Reference:
>
> [1] Abdulaal, A., Liu, Z., and Lancewicki, T. Practical approach to asynchronous multivariate time series anomaly detection and localization. In Proceedings of the 27th ACM SIGKDD conference on knowledge discovery & data mining, 2021.

---

> > ### Author Rebuttal · Reviewer_Rmqr · 2026-04-01
> >
> > Thank you for addressing all of my comments. Happy to update my score from weak accept to Accept.

---

> > > ### Author Response · Authors · 2026-04-01
> > >
> > > Thank you for your positive feedback and for updating your score! We really appreciate your support.

---

### Official Review · Reviewer_o1EU · 2026-03-13

**Soundness:** 3
**Presentation:** 4
**Significance:** 3
**Originality:** 3
**Overall Recommendation:** 5
**Confidence:** 3

**Summary:**

Time series analysis is indispensable in critical domains. However, accurately quantifying the value of temporal segments remains an important challenge. Practical applications further require valuation methods to possess learning-agnosticism and robustness to temporal distribution shifts, which further increases the complexity of the problem. To address this issue, this paper proposes the computational framework TimeLAVA based on the Selective Wavelet-based Wasserstein (WSW) discrepancy. This framework innovatively combines multi-scale wavelet transform with unbalanced optimal transport, and quantifies segment values through sensitivity analysis of the WSW dual formulation, ultimately aggregating segment-level scores into fine-grained point-level valuations. Experimental results demonstrate that TimeLAVA consistently outperforms existing state-of-the-art baseline methods in anomaly detection, data pruning/selection, and temporal label noise detection, and can improve the performance of downstream tasks.

**Compliance With Llm Reviewing Policy:**

Affirmed.

**Final Justification:**

After reviewing authors' replies to both myself and other reviewers, I  raise my score to 5. However, I strongly encourage the authors to carefully incorporate all feedback holistically to significantly improve the quality of the final manuscript.

**Key Questions For Authors:**

1.The performance of TimeLAVA proposed in this paper depends on the representativeness of the reference set. A biased or incomplete reference set may undervalue rare but important patterns. Can the authors propose mitigation strategies for this limitation and provide theoretical guarantees?
2.In the wavelet transform, the authors used the db4 wavelet and 2-level decomposition in all experiments. Can a detailed explanation be provided for the general applicability of this decision? If there is no general applicability, can the authors discuss how to select the wavelet basis and its decomposition level for TimeLAVA when dealing with different types of time series?
3.This paper only evaluated the performance of TimeLAVA on three tasks: anomaly detection, data pruning, and label noise detection. Can the authors discuss the applicability of this framework in other tasks, and whether adjustments to the framework are required according to different task requirements?

**Limitations:**

1.The valuation performance of TimeLAVA is entirely dependent on the representativeness of the reference set. However, datasets obtained in real-life scenarios are often biased or lack fault cases with ultra-low probabilities, meaning the acquired datasets are frequently incomplete and poorly representative. Furthermore, a fixed reference set cannot adapt to distribution shifts, and it is necessary to periodically maintain and collect representative reference sets to ensure the method continues to function effectively. This undoubtedly increases the operational costs in practical applications.

2.The interpretability of the valuation is poor. The method can only output the value score of a segment but struggles to explain why the segment possesses such a value. This limits its application in fields requiring high reliability and regulatory compliance.

3.If the selected reference set is biased, or if data from a certain group is insufficient due to practical reasons, the valuation results of TimeLAVA will perpetuate or even amplify such biases. This is more prominent in the healthcare field: biases in the reference set may lead to the underestimation of behavioral patterns of specific groups, triggering discrimination issues. The paper does not discuss such fairness risks nor propose corresponding mitigation strategies.

**Strengths And Weaknesses:**

Strengths:
S1: This paper provides a rational analysis of why multi-scale wavelet transform is integrated with Unbalanced Optimal Transport (UOT) to form the WSW discrepancy in TimeLAVA. To establish a complete and rigorous framework, the authors also present comprehensive mathematical justifications. Specifically, they prove that the defined wavelet distance is a valid metric. Additionally, full and detailed proofs are provided for two key theorems: the bounded sensitivity of the proposed WSW discrepancy to outlier contamination, and its connection to learning-agnostic generalization.

S2. The paper has a clear and reasonable structure. Starting from the dilemmas of existing methods and practical needs, it analyzes the conflicts and contradictions between core requirements and technical shortcomings in different domains layer by layer, and proposes targeted solutions. Overall, the content progresses logically, helping readers quickly grasp the innovations of TimeLAVA and enabling them to understand the practical dilemmas solved by the method.

Weaknesses:
W1: However, several critical issues undermine the overall soundness of the method. First, although the paper claims that TimeLAVA supports multivariate time series, the adopted strategy of "processing each channel independently and then aggregating" neglects the potential cross-channel temporal dependencies among variables. Second, the authors fix the wavelet basis as db4 with 2-level decomposition throughout the experiments without sufficient justification. This raises concerns that the performance gains might merely stem from hyperparameter tuning tailored to specific datasets. Finally, the benchmark datasets used in the experiments are relatively small in scale. For sequences with millions of time points, the computation of the pairwise cost matrix becomes infeasible. In the absence of approximate algorithms or scalability tests, the practical utility of this method in real-world big data scenarios is questionable.

---

> ### Author Rebuttal · Authors · 2026-03-29
>
> We sincerely thank the reviewer for the positive assessment and thoughtful feedback! Below we address each question and concern.
>
> > Q1, L1, L3: Reference set bias mitigation
>
> *Current mitigation, intrinsic robustness via UOT*: TimeLAVA's UOT partially addresses reference set bias. Unlike standard OT which forces all segments to match (producing false positives when the reference is biased), UOT allows segments to remain "unmatched" when the reference lacks certain patterns, so these segments receive correspondingly low matching weights rather than forced alignment to irrelevant samples.
>
> *Future directions for further mitigation*: We acknowledge that reference set bias remains an open challenge for all reference-based valuation methods. To further address this, a few directions seem worth exploring for future work:
> - Constructing multiple reference sets via repeated sampling from available data, averaging valuation results to reduce single-reference-set bias.
> - Using diversity-aware sampling (e.g., k-medoids) to maximize reference set coverage in the wavelet feature space.
>
> These strategies would also help mitigate the fairness concerns raised in L3, as a more representative reference set reduces the risk of systematically undervaluing certain groups. We will include these mitigation strategies in the Future Work section of the revised manuscript.
>
> > Q2, W2: Generality of db4 + 2-level decomposition
>
> We achieved consistent performance using the same configuration across diverse datasets spanning healthcare, finance, industrial IoT, and traffic domains, which strongly supports generality. db4 provides a strong balance between time localization and frequency resolution; 2 levels yield 3 frequency bands sufficient to cover the primary temporal scales. We chose this standard configuration for simplicity and consistency.
>
> That said, for specialized applications, the following guidance may help:
> - High-frequency signals (spikes, transients): 1-2 levels typically sufficient, as high-frequency details are captured in the shallowest detail coefficients.
> - Low-frequency / slowly-varying data: consider deeper decomposition (3-4 levels) to separate low-frequency trends from mid-frequency variations.
> - Very smooth signals: higher-order wavelets (db6, db8) may be preferable.
> - Discontinuous/step signals: lower-order wavelets (Haar, db2) may be more appropriate.
>
> We will include this practical guidance in the revised manuscript.
>
> > Q3: Applicability to other tasks
>
> TimeLAVA's framework naturally extends to additional tasks without structural modifications:
>
> 1. *Data valuation for time series classification*: Directly applicable by setting c>0 to leverage label information, assigning value scores to training samples to identify the most valuable/harmful samples for classifier training.
>
> 2. *Active learning*: TimeLAVA's valuation scores can guide annotation priorities by prioritizing samples with the largest distributional discrepancy from the current reference set to maximize information gain.
>
> These extensions require only task-appropriate choices of reference set and parameter c. We will discuss these potential applications in the revised manuscript.
>
> > W1: Cross-channel dependencies
>
> We acknowledge that independent per-channel processing does not explicitly model cross-channel dependencies. We adopt this design for simplicity and parallelizability: introducing cross-channel modeling (e.g., multivariate wavelet transforms or channel attention) would introduce model parameters or prior assumptions, conflicting with our learning-agnostic principle. Despite this, TimeLAVA achieves consistent performance on multivariate datasets, demonstrating the aggregation strategy's practical effectiveness. We will discuss this limitation explicitly in the revision and explore cross-channel modeling as future work.
>
> > W3: Scalability
>
> Our largest benchmark, KDD-CUP99, contains 494k time points, and TimeLAVA completes in ~100 seconds per dimension (Table 5). For even larger-scale data, one possible approximation strategy is to adopt a mini-batch approach similar to SAVA [1] to improve scalability. We will discuss this scalability strategy in the revision.
>
> > L2: Interpretability
>
> TimeLAVA actually offers more interpretability than purely score-based methods. The wavelet decomposition reveals which frequency bands contribute most to a segment's value, and the UOT transport plan shows which reference segments it matches (or fails to match). We agree that further work on user-facing explanations would be valuable and will discuss this in the revision.
>
> We are grateful for these constructive suggestions and will incorporate them in the revision! We hope our responses adequately address all concerns.
>
> Reference:
>
> [1] Kessler, S., Le, T., and Nguyen, V. SAVA: Scalable learning-agnostic data valuation. ICLR, 2025.

---

> > ### Author Rebuttal · Reviewer_o1EU · 2026-04-02
> >
> > Thank you for your detailed response. After reviewing your replies to both myself and other reviewers, I will raise my score to 5. However, I strongly encourage you to carefully incorporate all feedback holistically to further improve the quality of the final manuscript.

---

> > > ### Author Response · Authors · 2026-04-02
> > >
> > > Thank you for your follow-up and for raising your score! We also appreciate you taking the time to consider the feedback from other reviewers. We will incorporate all feedback to further improve the quality of the revised manuscript. Thank you again for your time and support.

---

### Decision · Program_Chairs · 2026-04-30

**Decision:**

Accept (regular)

**Comment:**

At the end of the discussion phase, three reviewers gave a recommendation of Accept and one reviewer gave a recommendation of Weak Reject. The dissenting reviewer's concern post-rebuttal revolved around whether this work's contribution was sufficiently novel. The authors argued that their novel technical content comes from how they connect existing components through novel theorems that are not trivial implications from prior theory. Other reviewers have indicated that the formulation and contributions are original.

During the rebuttal process, the authors included several new experiments and have promised to make several improvements to the manuscript based off of the thoughtful feedback from all reviewers. I urge the authors to follow through on these changes and expect to see this reflected in the camera-ready version of the paper.

I recommend that this work be accepted.